# HIERARCHICAL CLUSTERING FOR CONDITIONAL DIFFUSION IN IMAGE GENERATION

## ABSTRACT

Finding clusters of data points with similar characteristics and generating new cluster-specific samples can significantly enhance our understanding of complex data distributions. While clustering has been widely explored using Variational Autoencoders, these models often lack generation quality in real-world datasets. This paper addresses this gap by introducing TreeDiffusion, a deep generative model that conditions Diffusion Models on hierarchical clusters to obtain high-quality, cluster-specific generations. The proposed pipeline consists of two steps: a VAE-based clustering model that learns the hierarchical structure of the data, and a conditional diffusion model that generates realistic images for each cluster. We propose this two-stage process to ensure that the generated samples remain representative of their respective clusters and enhance image fidelity to the level of diffusion models. A key strength of our method is its ability to create images for each cluster, providing better visualization of the learned representations by the clustering model, as demonstrated through qualitative results. This method effectively addresses the generative limitations of VAE-based approaches while preserving their clustering performance. Empirically, we demonstrate that conditioning diffusion models on hierarchical clusters significantly enhances generative performance, thereby advancing the state of generative clustering models.

## 1 INTRODUCTION

Generative modeling and clustering are two fundamental yet distinct tasks in machine learning. Generative modeling focuses on approximating the underlying data distribution, enabling the generation of new samples (Kingma & Welling, 2014; Goodfellow et al., 2014). Clustering, on the other hand, seeks to uncover meaningful and interpretable structures within data through the unsupervised detection of intrinsic relationships and dependencies (Ward Jr, 1963; Ezugwu et al., 2022), facilitating better data visualization and interpretation. TreeVAE (Manduchi et al., 2023) was recently proposed to bridge these two research directions by integrating hierarchical dependencies into a deep latent variable model. While TreeVAE is effective at hierarchical clustering, it falls short in generating high-quality images. Like other VAE-based models, TreeVAE faces common issues such as producing blurry outputs (Bredell et al., 2023). In contrast, diffusion models (Sohl-Dickstein et al., 2015; Ho et al., 2020) have recently gained prominence for their superior image generation capabilities, progressively refining noisy inputs to produce sharp, realistic images.

Our work bridges this gap by introducing a second-stage diffusion model conditioned on cluster-specific representations learned by TreeVAE. The proposed framework, **TreeDiffusion**, combines the strengths of both models to generate high-quality, cluster-specific images, achieving strong performance in both clustering and image generation. The generative process begins by sampling the root embedding of a latent tree, which is learned during training. From there, the sample is propagated from the root to the leaf by (a) sampling a path through the tree and (b) applying a sequence of stochastic transformations to the root embedding along the chosen hierarchical path. Subsequently, the diffusion model leverages the hierarchical information by conditioning its reverse diffusion process on the sampled path representation of the latent tree. A key strength of TreeDiffusion is its ability to generate images for each cluster, providing enhanced visualization of the learned representations, as demonstrated by our qualitative results. The method produces leaf-specific images that share common general properties but differ by cluster-specific features, as encoded in the latent

hierarchy. This approach overcomes the generative limitations of VAE-based clustering models like TreeVAE while preserving their clustering performance.

**Our key contributions** include: (i) a unified framework that integrates hierarchical clustering into diffusion models, and (ii) a novel mechanism for controlling image synthesis. We demonstrate that our approach (a) surpasses the generative limitations of VAE-based clustering models, and (b) produces samples that are both more representative of their respective clusters and closer to the true data distribution.

## 2 RELATED WORK

**Variational Approaches for Hierarchical Clustering**   Since their introduction, Variational Autoencoders (Kingma & Welling, 2014, VAEs) have been often employed for clustering tasks, as they are particularly effective in learning structured latent representations of data (Jiang et al., 2017). Goyal et al. (2017), for example, integrates hierarchical Bayesian non-parametric priors to the latent space of VAEs by applying the nested Chinese Restaurant Processes to cluster the data based on infinitely deep and branching trees. Additionally, hierarchical clustering has been achieved through models such as DeepECT (Mautz et al., 2020) and TreeVAE (Manduchi et al., 2023), both of which grow and learn hierarchical representations during training. While DeepECT aggregates data into a hierarchical tree in a single shared latent space, TreeVAE learns a tree structure posterior distribution of latent stochastic variables. That is, TreeVAE models the data distribution by learning an optimal tree structure of latent variables, resulting in latent embeddings that are automatically organized into a hierarchy, mimicking the hierarchical clustering process. Single-cell TreeVAE (Vandenhirtz et al., 2024, scTree) further extends TreeVAE to cluster single-cell RNA sequencing data by integrating batch correction, facilitating biologically plausible hierarchical structures. Although the aforementioned works have proven effective for clustering, their generative capabilities often fall short, with few offering quantitative or qualitative evaluations of their generative model performance.

**Diffusion Models**   Diffusion models have largely become state-of-the-art for image generation tasks (Sohl-Dickstein et al., 2015; Ho et al., 2020). Nichol & Dhariwal (2021) and Dhariwal & Nichol (2021) introduced enhancements to the architecture and training procedures of the Denoising Diffusion Probabilistic Model (DDPM). Meanwhile, other works, such as Song et al. (2020) and Salimans & Ho (2022), have made significant strides in reducing the sampling times for diffusion models, addressing one of their primary drawbacks. Song et al. (2023) further introduced consistency models, a new family of models that generate high-quality samples by directly mapping noise to data, enabling fast one-step generation while still allowing multistep sampling to balance computation and sample quality. Moreover, Song & Ermon (2019) proposed an alternative formulation of diffusion modeling through their core-based generative model, known as the noise conditional score network (NCSN). Finally, recent work has further pushed the boundaries of diffusion models by relocating the diffusion process to the latent spaces of autoencoders, as demonstrated in works like LSGM (Vahdat et al., 2021) and Stable Diffusion (Rombach et al., 2022).

One drawback of diffusion models is that their latent variables lack interpretability compared to the latent spaces of VAEs. To leverage the strengths of both approaches, researchers have begun developing architectures that combine the more interpretable latent spaces of VAEs with the advanced generative capabilities of diffusion models. Notable examples include DiffuseVAE (Pandey et al., 2022), Diffusion Autoencoders (Preechakul et al., 2022), and InfoDiffusion (Wang et al., 2023). Representation-Conditioned image Generation (Li et al., 2023) illustrates how self-supervised learning can improve generative diffusion frameworks in unsupervised settings, reducing the gap between class-conditional and unconditional image generation.

**Connecting Diffusion with Clustering**   The research most closely related to our work focuses on using clustering as conditioning signals for diffusion models to enhance their generative quality. For instance, Adaloglou et al. (2024) propose an approach that utilizes cluster assignments from k-means or TEMI clustering (Adaloglou et al., 2023). Similarly, Hu et al. (2023) introduces a framework that employs the k-means clustering algorithm as an annotation function, generating self-annotated image-level, box-level, and pixel-level guidance signals. Both studies demonstrate the benefits of conditioning on clustering information to improve generative performance without going into the specifics of clustering performance itself. In contrast, our research further investigates

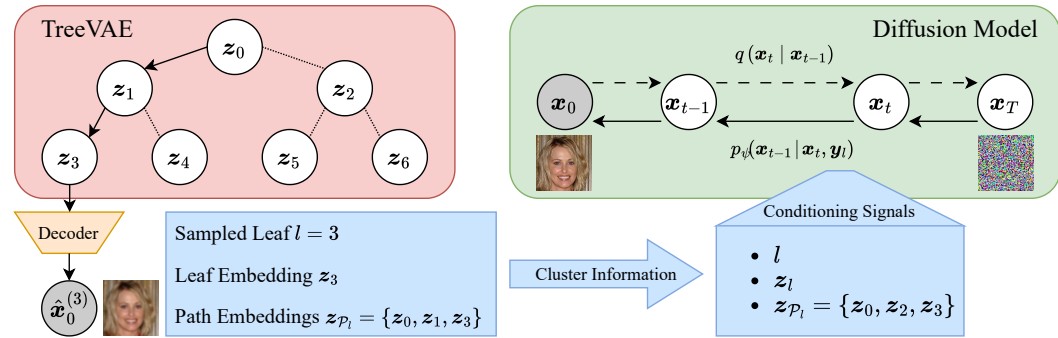

Figure 1: Schematic overview of the TreeDiffusion framework: TreeVAE encodes data into hierarchical latent variables. A path is sampled through the tree, ending at a leaf, with the leaf embedding decoded to generate a reconstruction. The diffusion model leverages the cluster information from TreeVAE by conditioning its reverse process on the sampled leaf and the path embeddings, producing a sharper cluster-specific version of the image.

which types of clustering information are most beneficial for the model, employing learned latent cluster representations alongside cluster assignments for conditioning. Related to conditioning on clusters, both kNN-Diffusion (Sheynin et al., 2023) and Retrieval-Augmented Diffusion Models (Blattmann et al., 2022) utilize nearest neighbor retrieval to condition generative models on similar embeddings, minimizing the need for large parametric models and paired datasets in tasks like text-to-image synthesis. Diffusion models have also been applied in incomplete multiview clustering to generate missing views to improve clustering performance, as demonstrated by (Wen et al., 2024; 2020). Recent works (Liu et al., 2023; Su et al., 2024) analyze the capability of diffusion models for unsupervised concept discovery, wherein image datasets are decomposed into meaningful compositional representations, similar to clustering. On a different note, (Wang et al., 2024) shows that training diffusion models is equivalent to solving a subspace clustering problem, explaining their ability to learn image distributions with few samples. Additionally, Palumbo et al. (2023) employ post-hoc diffusion models to enhance the generation quality of their multimodal clustering models. However, to the best of our knowledge, there is currently no diffusion model that leverages the hierarchical structure of the data to enhance the interpretability and generative performance of generative clustering models.

## 3 METHOD

We propose **TreeDiffusion**[1], a two-stage framework consisting of a first-stage VAE-based generative hierarchical clustering model, followed by a second-stage hierarchy-conditional diffusion model. This novel combination of VAEs and diffusion models extends the DiffuseVAE framework introduced by Pandey et al. (2022) to hierarchical clustering. It enables cluster-guided diffusion in unsupervised settings, as opposed to classifier-guided diffusion for labeled data, as introduced by Dhariwal & Nichol (2021). In our framework, TreeVAE (Manduchi et al., 2023) serves as the clustering model, encoding hierarchical clusters within its latent tree structure, where the leaves represent the clusters. A denoising diffusion implicit model (DDIM) (Song et al., 2020), conditioned on the TreeVAE leaves, utilizes these leaf representations to generate improved cluster-conditional samples. Figure 1 illustrates the workflow of TreeDiffusion.

### 3.1 HIERARCHICAL CLUSTERING WITH TREEVAE

The first part of TreeDiffusion involves an adapted version of the Tree Variational Autoencoder (TreeVAE) by Manduchi et al. (2023). TreeVAE is a generative model that learns to hierarchically separate data into clusters via a latent tree structure. During training, the model dynamically grows a binary tree structure of stochastic variables, $\mathcal{T}$. The process begins with a tree composed of a root node and two child nodes and it optimizes the corresponding ELBO over a fixed number of

---

[1]The code will be published upon acceptance.

epochs. Afterward, the tree expands by adding two child nodes to an existing leaf node, prioritizing nodes with the highest assigned sample count to promote balanced leaves. This expansion continues iteratively, training only the subtree formed by the new leaves while freezing the rest of the model. This process repeats until the tree reaches a predefined depth or leaf count, alternating between optimizing model parameters and expanding the tree structure.

Let the set $\mathbb{V}$ represent the nodes of the tree. Each node corresponds to a stochastic latent variable, denoted as $\mathbf{z}_0, \ldots, \mathbf{z}_V$. Each latent variable follows a Gaussian distribution, whose parameters depend on their parent nodes through neural networks called *transformations*. The set of leaves $\mathbb{L}$, where $\mathbb{L} \subset \mathbb{V}$, represents the clusters present in the data. Starting from the root node, $\mathbf{z}_0$, a given sample traverses the tree to a leaf node, $\mathbf{z}_l$, in a probabilistic manner. The probabilities of moving to the left or right child at each internal node are determined by neural networks termed *routers*. These decisions, denoted by $c_i$ for each non-leaf node $i$, follow a Bernoulli distribution, where $c_i = 0$ indicates the selection of the left child. The path $\mathcal{P}_l$ refers to the sequence of nodes from the root to a leaf $l$. Thus, the latent tree encodes a sample-specific probability distribution of paths. Each leaf embedding, $\mathbf{z}_l$ for $l \in \mathbb{L}$, represents the learned data representations, and leaf-specific decoders use these embeddings to reconstruct or generate new cluster-specific images, i.e. given a dataset $\boldsymbol{X}$, TreeVAE reconstructs $\hat{\boldsymbol{X}} = \{ \hat{\boldsymbol{X}}^{(l)} \mid l \in \mathbb{L} \}$. The generative model (1) and inference model (2) of TreeVAE are defined as follows:

$$p_\theta \left( \boldsymbol{z}_{\mathcal{P}_l}, \mathcal{P}_l \right) = p\left( \boldsymbol{z}_0 \right) \prod_{i \in \mathcal{P}_l \backslash \{0\}} \underbrace{p\left( \mathrm{c}_{pa(i) \to i} \mid \boldsymbol{z}_{pa(i)} \right)}_{\text{decision probability}} \underbrace{p\left( \boldsymbol{z}_i \mid \boldsymbol{z}_{pa(i)} \right)}_{\text{sample probability}} \tag{1}$$

$$q \left( \boldsymbol{z}_{\mathcal{P}_l}, \mathcal{P}_l \mid \boldsymbol{x} \right) = q\left( \boldsymbol{z}_0 \mid \boldsymbol{x} \right) \prod_{i \in \mathcal{P}_l \backslash \{0\}} q\left( \mathrm{c}_{pa(i) \to i} \mid \boldsymbol{x} \right) q\left( \boldsymbol{z}_i \mid \boldsymbol{z}_{pa(i)} \right) \tag{2}$$

The objective of the model is to maximize the evidence lower bound (ELBO), which consists of two main components: the reconstruction term $\mathcal{L}_{rec}$ and the KL divergence term, which is broken down into contributions from the root node, internal nodes, and decision probabilities:

$$\mathcal{L}(\boldsymbol{x} \mid \mathcal{T}) := \underbrace{\mathbb{E}_{q(\boldsymbol{z}_{\mathcal{P}_l}, \mathcal{P}_l \mid \boldsymbol{x})}[\log p(\boldsymbol{x} \mid \boldsymbol{z}_{\mathcal{P}_l}, \mathcal{P}_l)]}_{\mathcal{L}_{rec}} - \underbrace{\mathrm{KL}(q(\boldsymbol{z}_{\mathcal{P}_l}, \mathcal{P}_l \mid \boldsymbol{x}) \| p(\boldsymbol{z}_{\mathcal{P}_l}, \mathcal{P}_l))}_{\mathrm{KL}_{root} + \mathrm{KL}_{nodes} + \mathrm{KL}_{decisions}}. \tag{3}$$

In this work, we modify the architectural design of TreeVAE, which originally uses an encoder to project images into flattened representations and relies on MLP layers for subsequent processing. Instead, we utilize convolutional layers throughout the model, which leverage lower-dimensional, multi-channel representations, thereby avoiding flattening the representations. Additionally, we incorporate residual connections to enhance the training stability and model performance. These modifications aim to preserve spatial information and facilitate more efficient learning, making the model particularly effective for image data. Nevertheless, it is important to note that this model encounters the common VAE issue of producing blurry image generations (Bredell et al., 2023). Despite this limitation, the reconstructed images and learned hierarchical clustering still offer meaningful representations of the data, which are used in the second stage of the proposed TreeDiffusion framework.

## 3.2 CLUSTER-CONDITIONED DIFFUSION

The second part of TreeDiffusion incorporates a conditional diffusion model. Based on the learned latent tree, the first-stage TreeVAE generates hierarchical clusters, which guide the second-stage diffusion process. The diffusion process involves two processes: forward noising and reverse denoising. We assume the same forward process as in standard Denoising Diffusion Probabilistic Models (Ho et al., 2020, DDPM), which gradually introduces noise to the data $\boldsymbol{x}_0$ over $T$ steps. The intermediate states, $\boldsymbol{x}_t$ for $t = 1, \ldots, T$, follow a trajectory determined by a noise schedule $\beta_1, \ldots, \beta_T$ that controls the rate of data degradation:

$$q \left( \boldsymbol{x}_{1:T} \mid \boldsymbol{x}_0 \right) = \prod_{t=1}^{T} q\left( \boldsymbol{x}_t \mid \boldsymbol{x}_{t-1} \right) \tag{4}$$

$$q \left( \boldsymbol{x}_t \mid \boldsymbol{x}_{t-1} \right) = \mathcal{N} \left( \sqrt{1 - \beta_t} \boldsymbol{x}_{t-1}, \beta_t \boldsymbol{I} \right) \tag{5}$$

$$q \left( \boldsymbol{x}_t \mid \boldsymbol{x}_0 \right) = \mathcal{N} \left( \sqrt{\bar{\alpha}_t} \boldsymbol{x}_0, (1 - \bar{\alpha}_t) \boldsymbol{I} \right), \text{ where } \alpha_t = (1 - \beta_t) \text{ and } \bar{\alpha}_t = \prod_{s=1}^{t} \alpha_s. \tag{6}$$

For the reverse process, we modify the DiffuseVAE framework by Pandey et al. (2022), where a VAE generates the initial, typically blurred images, and a diffusion model refines them to produce sharper, higher-quality outputs. Instead of starting the denoising process with VAE reconstructions, our model begins with random noise. Unlike DiffuseVAE, TreeDiffusion conditions exclusively on the latent information provided by TreeVAE, denoted as $\boldsymbol{y}_l$. The tree leaf $l$ represents the chosen cluster and is selected by sampling from the TreeVAE path probabilities. For the cluster-specific conditioning information $\boldsymbol{y}_l$, we considered:

$$\boldsymbol{y}_l = \begin{cases} l & \text{leaf assignment,} \\ \boldsymbol{z}_l & \text{leaf embedding,} \\ \boldsymbol{z}_{\mathcal{P}_l} & \text{set of latent embeddings from root to leaf.} \end{cases} \quad (7)$$

The conditioning information $\boldsymbol{y}_l$ guides the U-Net decoder (Ronneberger et al., 2015; Nichol & Dhariwal, 2021) throughout the denoising process. For $\boldsymbol{y}_l = l$ or $\boldsymbol{y}_l = \boldsymbol{z}_l$, the conditioning signal is directly projected to the same dimensionality as the time-step embeddings using a block that consists of two MLP layers with a SiLU activation in between. In contrast, when using the set of latent embeddings from the root to the leaf in the learned hierarchy provided by TreeVAE, i.e., $\boldsymbol{y}_l = \boldsymbol{z}_{\mathcal{P}_l}$, each node embedding and its corresponding node index are projected independently. Specifically, one projection block is used for the node indices and another for the node embeddings. These projected values are then aggregated to form a unified conditioning signal, which is subsequently added to the time-step embeddings in the U-Net. For the experiments in Section 4.1 and Section 4.2, we employ $\boldsymbol{y}_l = \boldsymbol{z}_{\mathcal{P}_l}$, as this configuration truly utilizes the hierarchical information provided by Tree-VAE, rather than just the flat cluster assignments. This approach empirically improves generative performance, as demonstrated in Section 4.3.

This conditioning mechanism directly influences the reverse process. Let $\psi$ denote the parameters of the denoising model, and let $p(l|\boldsymbol{x}_0)$ be the probability that the sample $\boldsymbol{x}_0$ is assigned to leaf $l$. The reverse process can then be summarized as follows:

$$\begin{aligned} l &\sim p(l|\boldsymbol{x}_0), \\ p_\psi(\boldsymbol{x}_{0:T} \,|\, \boldsymbol{y}_l) &= p\left(\boldsymbol{x}_T\right) \prod_{t=1}^{T} p_\psi(\boldsymbol{x}_{t-1} \,|\, \boldsymbol{x}_t, \boldsymbol{y}_l), \end{aligned} \quad (8)$$

This method ensures that leaves with smaller assignment probabilities are considered, encouraging the diffusion model to perform effectively across all leaves. Consequently, our approach addresses the distinct clusters inherent to TreeVAE, allowing the model to adapt to different clusters and encouraging cluster-specific refinements in the images. This guidance in the image generation process assists the denoising model in learning cluster-specific image reconstructions. Because of the large number of denoising steps required, DDPM sampling can be computationally expensive. To address this issue, we opt for the DDIM sampling procedure (Song et al., 2020) instead of the standard DDPM (Ho et al., 2020). DDIMs significantly accelerate inference by utilizing only a subset of denoising steps, making the process more efficient while maintaining high-quality results.

Finally, by employing a two-stage training strategy, where the conditional diffusion model is trained using a pre-trained TreeVAE model, TreeDiffusion preserves the hierarchical clustering performance of TreeVAE. Hence, we can combine the effective clustering of TreeVAE with the superior image generation capabilities of diffusion models.

## 4 EXPERIMENTS

We present a series of experiments designed to evaluate the performance of TreeDiffusion across various datasets. In Section 4.1, we compare the clustering and generative performance of TreeDiffusion to TreeVAE (Manduchi et al., 2023) using several benchmark datasets, including MNIST (Lecun et al., 1998), FashionMNIST (Xiao et al., 2017), CIFAR-10 (Krizhevsky, 2009), and CUBICC (Palumbo et al., 2023). The CUBICC dataset, a variant of the CUB Image-Captions dataset (Wah et al., 2011; Shi et al., 2019), contains images of birds grouped into eight specific species, allowing for a detailed analysis of clustering performance. Additionally, we conduct generative evaluations using the CelebA dataset (Liu et al., 2015) to assess the model's ability to generate high-quality images. In Section 4.2, we assess how cluster-specific the generated images are and analyze the variability among samples generated from the same cluster, examining whether there

are any indications of mode collapse. Finally, in Section 4.3, we perform an ablation study on the conditioning signals to compare the generative capabilities of different model configurations and identify the signals that most effectively enhance performance. Through these experiments, we aim to demonstrate the effectiveness of the TreeDiffusion model in both clustering and generation tasks.

## 4.1 GENERATIVE AND CLUSTERING PERFORMANCE

The following analysis compares two models: the TreeVAE and the proposed TreeDiffusion. The TreeDiffusion models are conditioned on the path $z_{\mathcal{P}_l}$ retrieved from the TreeVAE. Reconstruction performance is assessed using the Fréchet Inception Distance (Heusel et al., 2017, FID), calculated for the reconstructed images from the images in the test set. Additionally, we compute the FID score using $10,000$ newly generated images to evaluate generative performance. Clustering performance is measured using accuracy (ACC) and normalized mutual information (NMI). Table 1 presents the results of this analysis.

Table 1: Test set generative and clustering performances of different TreeVAE models. Means and standard deviations are computed across 10 runs with different seeds.

| Dataset | Method | FID (rec) ↓ | FID (gen) ↓ | ACC ↑ | NMI ↑ |
|---|---|---|---|---|---|
| MNIST | TreeVAE | $24.0 \pm 0.9$ | $21.8 \pm 0.7$ | $82.1 \pm 4.8$ | $82.8 \pm 3.1$ |
| | TreeDiffusion | $\mathbf{1.5} \pm 0.0$ | $\mathbf{1.8} \pm 0.1$ | | |
| Fashion | TreeVAE | $40.7 \pm 2.1$ | $41.9 \pm 2.1$ | $58.5 \pm 2.9$ | $62.6 \pm 2.5$ |
| | TreeDiffusion | $\mathbf{5.5} \pm 0.6$ | $\mathbf{5.4} \pm 0.4$ | | |
| CIFAR-10 | TreeVAE | $175.8 \pm 1.4$ | $188.0 \pm 2.0$ | $50.1 \pm 3.5$ | $41.0 \pm 2.7$ |
| | TreeDiffusion | $\mathbf{12.5} \pm 0.4$ | $\mathbf{17.8} \pm 0.4$ | | |
| CUBICC | TreeVAE | $232.5 \pm 7.1$ | $255.3 \pm 8.8$ | $40.1 \pm 1.8$ | $33.3 \pm 1.4$ |
| | TreeDiffusion | $\mathbf{13.4} \pm 0.9$ | $\mathbf{29.0} \pm 5.4$ | | |
| CelebA | TreeVAE | $75.2 \pm 15.0$ | $77.9 \pm 5.6$ | — | — |
| | TreeDiffusion | $\mathbf{14.1} \pm 6.0$ | $\mathbf{18.4} \pm 7.2$ | | |

Notably, the clustering performance remains identical for both models, as TreeDiffusion leverages the hierarchical clustering information from the pre-trained TreeVAE. The results indicate that TreeDiffusion significantly enhances the generative capabilities of the model, reducing FID scores by roughly an order of magnitude across all datasets. This improvement is particularly evident in the quality of the generated images, as illustrated in Figure 2 for the CUBICC dataset. Here, we visually compare the samples generated by TreeDiffusion with those produced by the underlying TreeVAE, which supplied the cluster path as the conditioning signal for TreeDiffusion. The Tree-VAE model continues to produce visibly blurry images, whereas TreeDiffusion generates noticeably sharper samples that adhere better to the data distribution.

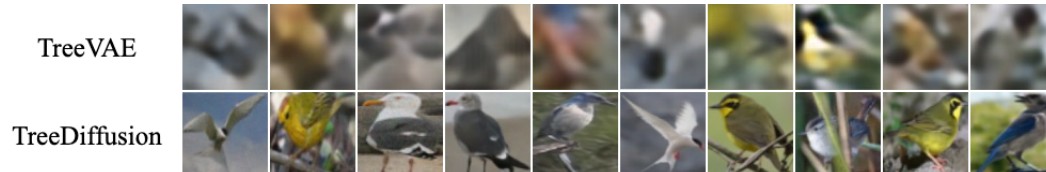

Figure 2: (Top) Ten different samples generated by the TreeVAE model, each generated by sampling one path in the tree. (Bottom) Corresponding samples from the TreeDiffusion model, conditioned on the same selected path and embeddings from TreeVAE.

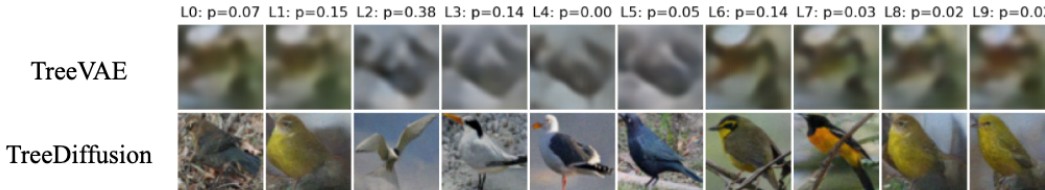

Figure 3: Image generations from every leaf of the (top) TreeVAE and (bottom) TreeDiffusion model, both trained on the CUBICC dataset. Each row shows the generated images from all leaves of the respective model, starting with the same root sample. The corresponding leaf probabilities are shown above each image and are identical across both models by design.

## 4.2 CLUSTER-SPECIFIC REPRESENTATIONS

**Higher quality cluster-specific generations**   In Figure 2, we present randomly generated images for the CUBICC dataset for both TreeVAE and TreeDiffusion, where each column corresponds to one generation process. For each generation, we first sample the root embedding; then, we sample the path in the tree and the refined representations along the selected path iteratively until a leaf is reached. The hierarchical representation is then used to condition the inference in TreeDiffusion. As can be seen, the TreeDiffusion generations show substantially higher generative quality. Additionally, we examine further the first generated sample from Figure 2. For this sample, we present the generations from all leaves in Figure 3 by propagating the corresponding root representation across all paths in the tree. Note that leaf "L2" has the highest path probability across all leaves. When comparing the generated images across the leaves for both models, it is evident that TreeDiffusion not only produces sharper images for all clusters but also generates a greater diversity of images. We ensured the same level of stochasticity for both models, eliminating potential confounding factors. Therefore, the observed diversity stems from the models themselves. As a result, the TreeDiffusion model can generate cluster-specific images that preserve the overall color and structure seen in TreeVAE images while significantly enhancing the distinctiveness and clarity of the images for each cluster. Further examples of the leaf-specific image generations can for TreeVAE and TreeDiffusion can be found in C.2.

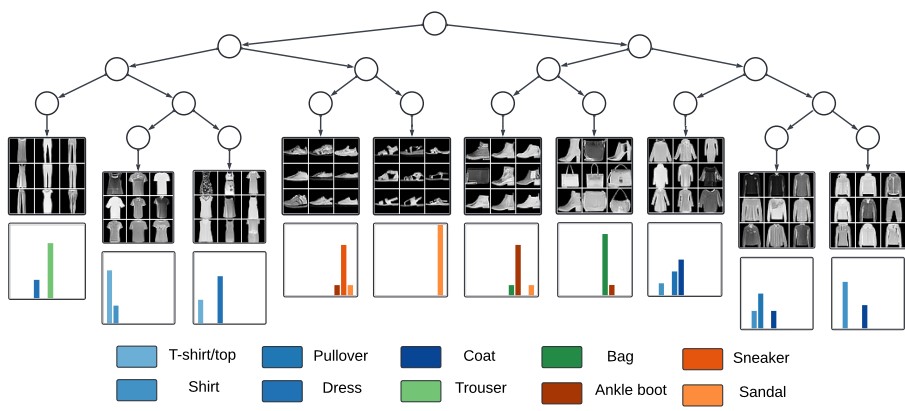

Figure 4: TreeDiffusion model trained on FashionMNIST. For each cluster, random newly generated images are displayed. Below each set of images, a normalized histogram (ranging from 0 to 1) shows the distribution of predicted classes from an independent, pre-trained classifier on FashionMNIST for all newly generated images in each leaf with a significant probability of reaching that leaf.

**Hierarchical information is retained across generations**   To assess whether the newly generated images retain their hierarchical information, we train a classifier on the original training datasets and then utilize it to classify the newly generated images from our TreeDiffusion. Specifically, we

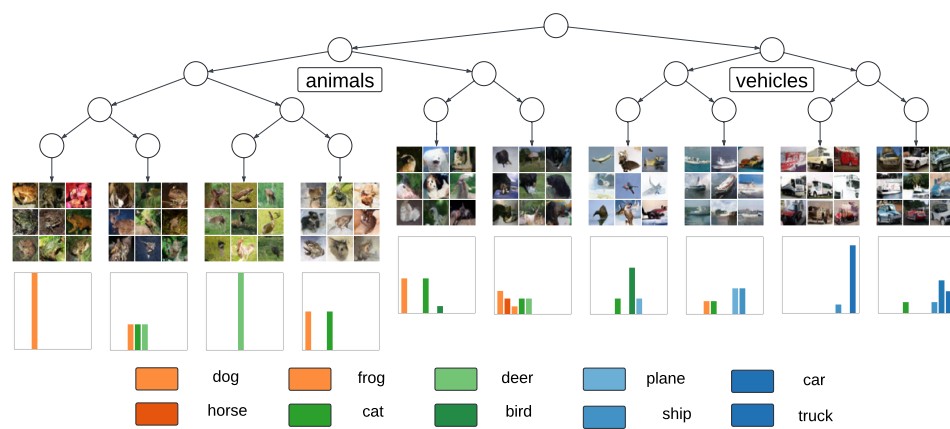

Figure 5: TreeDiffusion model trained on CIFAR-10. For each cluster, random newly generated images are displayed. Below each set of images, a normalized histogram (ranging from 0 to 1) shows the distribution of predicted classes from an independent, pre-trained classifier on CIFAR-10 for all newly generated images in each leaf with a significant probability of reaching that leaf.

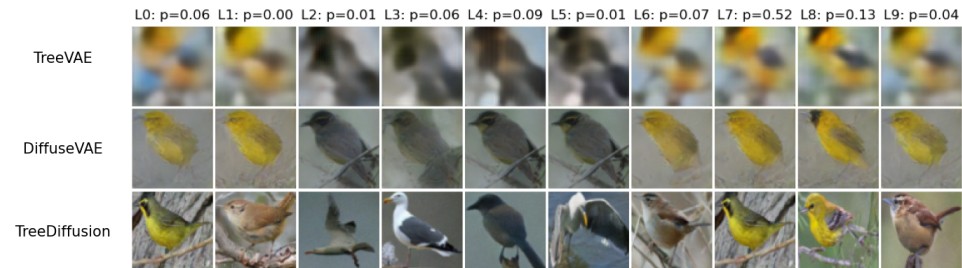

Figure 6: Image generations from each leaf of (top) a TreeVAE, (middle) a DiffuseVAE which only conditions on the reconstruction from the TreeVAE, and (bottom) a TreeDiffusion model conditioned on the path embeddings, all trained on CUBICC. Each row displays the generated images from all leaves of the specified model, starting with the same sample from the root. The corresponding leaf probabilities are shown at the top of the image and are, by design, the same for all models.

classify the newly generated images for each cluster separately. Ideally, "pure" leaves should be characterized by leaf generations that are classified into one or very few classes from the original dataset. For this classification task, we utilize a ResNet-50 model He et al. (2016) trained on each dataset. In Figure 4, we present randomly generated images from a TreeDiffusion model trained on FashionMNIST, together with normalized histograms depicting the distribution of the predicted classes for each leaf. For instance, clusters representing trousers and bags appear to accurately and distinctly capture their respective classes, as all their generated images are classified into one group only. Conversely, certain clusters are characterized by a mixture of classes, indicating that they are grouped together. Further results can be observed for the CIFAR-10 or MNIST dataset, shown in Figure 5 and Figure 8, respectively. Overall, we observe that the leaf-specific generations retain the hierarchical clustering structure found by TreeVAE, thereby enhancing interpretability in diffusion models.

**On the benefits of hierarchical conditioning** We hereby assess whether the conditioning on hierarchical representations improves cluster-specific generative quality. To this end, we compare the generations of TreeDiffusion, which is conditioned on the hierarchical representation, to a baseline, here defined as cluster-unconditional, that is conditioned only on the leaf reconstructions. For this experiment, we use the previously introduced independent classifier to create histograms for each leaf to evaluate how cluster-specific the newly generated images are.

As previously mentioned, ideally, the majority of generated images from one leaf should be classified into one or very few classes from the original dataset. To quantify this, we compute the average entropy for all leaf-specific histograms. Lower entropy indicates less variation in the histograms and, thus, more leaf-specific generations. Table 2 presents the results across all datasets. The conditional model consistently shows lower mean entropy, indicating that, for most datasets, conditioning on the hierarchy indeed helps guide the model to generate more distinct and representative images for each leaf.

Table 2: Cluster-specificity of TreeDiffusion generations for cluster-unconditional and cluster-conditional reverse models, measured by mean entropy. Lower entropy indicates more cluster-specific generations. The best result for each dataset is marked in **bold**.

| Dataset | Method | Mean Entropy |
|---------|--------|--------------|
| MNIST | unconditional | 1.24 |
|  | conditional | **0.33** |
| Fashion | unconditional | 0.66 |
|  | conditional | **0.65** |
| CIFAR10 | unconditional | 1.12 |
|  | conditional | **0.93** |
| CUBICC | unconditional | **0.07** |
|  | conditional | 0.20 |

However, for the CUBICC dataset, we observe that the mean entropy is lower for the cluster-unconditional model. This is because the classifier tends to predict all images into a single class, a result of model degeneration, where it primarily generates images for only a few classes. Figure 6 visually presents the leaf generations for one sample of these models alongside the underlying TreeVAE generations. It can be observed that both the cluster-unconditional and conditional models exhibit a significant improvement in image quality. However, the images in the cluster-conditional model are more diverse, demonstrating greater adaptability for each cluster. Notably, across all models, the leaf-specific images share common properties, such as background color and overall shape, sampled at the root while varying in cluster-specific features from leaf to leaf within each model.

## 4.3 ABLATION STUDY ON CONDITIONING INFORMATION

Finally, we perform an ablation study to assess the effects of various conditioning signals on the generative performance of the proposed approach. The results, outlined in Table 3, show the FID score calculated from $10,000$ samples generated using 100 DDIM steps, averaged over 10 random seeds. The findings indicate that utilizing information from the latent leaf — whether through leaf assignment, leaf embedding, or both — yields better generative performance compared to using only the leaf reconstruction. Additionally, conditioning on the full path $z_{\mathcal{P}_l}$, which incorporates all embeddings and intermediate node assignments from the root to the leaf, further enhances the performance, underscoring the effectiveness of hierarchical clustering information. Notably, conditioning on the latent path also exceeds the generative performance of a fully unconditional, vanilla DDIM model, which achieves an average FID of $18.1$. Consequently, utilizing the path $z_{\mathcal{P}_l}$ from the hierarchical structure not only results in more structured generations, as demonstrated in Figure 3, but can also enhance the generative performance of generative clustering models.

Table 3: Effect of conditioning signals on generative performance for CIFAR-10. FID scores for $10'000$ samples (lower is better) computed across 10 random model initializations.

| Leaf Reconstruction $\hat{x}_0^{(l)}$ | Leaf Assignment $l$ | Leaf Embedding $z_l$ | Path $z_{\mathcal{P}_l}$ | FID $\downarrow$ |
|:---:|:---:|:---:|:---:|:---|
| ✓ |  |  |  | $19.7 \pm 0.2$ |
| ✓ | ✓ |  |  | $19.1 \pm 0.3$ |
| ✓ |  | ✓ |  | $18.9 \pm 0.3$ |
| ✓ | ✓ | ✓ |  | $19.2 \pm 0.2$ |
|  | ✓ | ✓ |  | $19.1 \pm 0.5$ |
| ✓ |  |  | ✓ | $18.2 \pm 0.3$ |
|  |  |  | ✓ | $\mathbf{17.8} \pm 0.4$ |
| Vanilla DDIM |  |  |  | $18.1 \pm 0.3$ |

## 5 CONCLUSION

In this work, we present TreeDiffusion, a novel approach to integrate hierarchical clustering into diffusion models. By enhancing TreeVAE with a Denoising Diffusion Implicit Model conditioned on cluster-specific representations, we propose a model capable of generating distinct, high-quality images that faithfully represent their respective data clusters. This approach not only improves the visual fidelity of generated images but also ensures that these representations are true to the underlying data distribution. TreeDiffusion offers a robust framework that bridges the gap between clustering precision and generative performance, thereby expanding the potential applications of generative models in areas requiring detailed and accurate visual data interpretation.

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

# A  DETAILED FORMULATION OF TREEVAE

The TreeVAE is a novel deep generative model proposed by Manduchi et al. (2023). Built upon the VAE framework originally introduced by Kingma & Welling (2014), TreeVAEs inherently possess stochastic latent variables. However, unlike traditional VAEs, TreeVAEs organize these latent variables in a learnable binary tree structure , enabling them to capture complex hierarchical relationships.

In the TreeVAE framework, the tree hierarchically divides the data, yielding separate stochastic embeddings for each node. This division into nodes induces a probability distribution, allowing each sample to navigate through the nodes of the tree in a probabilistic manner, from the root to the leaf. Ideally, high-variance features should be partitioned at earlier stages in the tree, while deeper nodes encapsulate more detailed concepts. The flexible tree structure is learned during training and is specific to the data distribution. This structure allows the incorporation of sample-specific probability distributions over the different paths in the tree, as further explained in Section A.1. In this manner, TreeVAEs contribute to the generation of comprehensive hierarchical data representations.

TreeVAE combines elements from both LadderVAE (Sønderby et al., 2016) and ANT (Tanno et al., 2019), aiming to create a VAE-based model capable of performing hierarchical clustering via the latent variables. Similar to LadderVAE, the inference and the generative model share the same top-down hierarchical structure. In fact, if we isolate a path from the root to any leaf in the tree structure of TreeVAE, we obtain an instance of the top-down model seen in LadderVAE. On the other side, both Adaptive Neural Trees and TreeVAE engage in representation and architecture learning during training. However, while ANTs are tailored for regression and classification tasks, TreeVAE's focus lies in hierarchical clustering and generative modeling. This allows TreeVAEs to generate new class-specific data samples based on the latent embeddings of the leaves from the tree.

## A.1  MODEL FORMULATION

TreeVAE comprises an inference model and a generative model, which share the global structure of the binary tree $\mathcal{T}$. Following Manduchi et al. (2023), the tree structure is learned during training using dataset $\boldsymbol{X}$ and a predetermined maximum tree depth denoted as $H$. Specifically, the tree structure entails the set of nodes $\mathbb{V} = \{0, ..., V\}$, the subset of leaves $\mathbb{L} \subset \mathbb{V}$, and the set of edges $\mathcal{E}$.

While the global structure of the tree is the same for all samples in the data, the latent embeddings $\boldsymbol{z} = \{\boldsymbol{z}_0, ..., \boldsymbol{z}_V\}$ for all nodes and the so-called decisions $\boldsymbol{c} = \{c_0, ..., c_{V-|\mathbb{L}|}\}$ for all non-leaf nodes are unique to each sample, representing sample-specific random variables. The latent embeddings $\boldsymbol{z}$ are modeled as Gaussian random variables. Their distribution parameters are determined by functions that depend on the latent embeddings of the parent node. These functions, referred to as "transformations", are implemented as MLP. Moreover, each decision variable $c_i$ corresponds to a Bernoulli random variable, where $c_i = 0$ signifies the selection of the left child at internal node $i$. These decision variables influence the traversal path within the tree during both the generative and inference processes. The parameters governing the Bernoulli distributions are functions of the respective node value, parametrized by MLP termed "routers".

The transformations and routers are learned during the model training process, as further elaborated in Section A.2, which comprehensively covers all aspects of model training. The exact parametrizations of the transformations and routers vary depending on whether they are applied in the context of the inference or generative model. Subsequently, the specifics of both of these models will be explored.

### GENERATIVE MODEL

The top-down generative model of TreeVAE, illustrated in Figure 7 on the right side, governs the generation of new samples. It starts by sampling a latent representation $\boldsymbol{z}_0$ from a standard Gaussian distribution for the root node, serving as the initial point for the generation process:

$$p_\theta\left(\boldsymbol{z}_0\right) = \mathcal{N}\left(\boldsymbol{z}_0 \mid \boldsymbol{0}, \boldsymbol{I}\right). \tag{9}$$

This latent embedding then traverses through the tree structure, leading to new samples being generated at each node based on their ancestor nodes. These remaining nodes in the tree are characterized

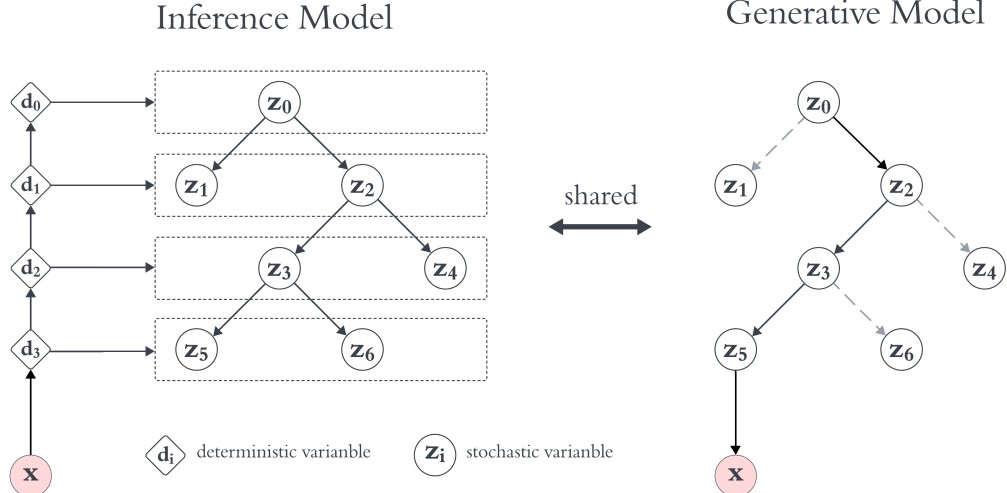

Figure 7: Illustration of the TreeVAE model structure. The learned tree topology is shared between the inference and generative models.

by their own latent representations, which are sampled conditionally on the sample-specific embeddings of their parent nodes. This process can be mathematically expressed as follows:

$$p\left(z_i \mid z_{pa(i)}\right) = \mathcal{N}\left(z_i \mid \mu_{p,i}(z_{pa(i)}), \sigma_{p,i}^2(z_{pa(i)})\right), \tag{10}$$

where $\{\mu_{p,i}, \sigma_{p,i} \mid i \in \mathbb{V}\backslash\{0\}\}$ correspond to the transformation neural networks associated with the generative model, as denoted by the subscript $p$.

Following the generation of latent representations for each node $i$ in the tree, the decision regarding traversal to the left or right child node are determined based on the sampled $z_i$ value. These decisions, denoted by $c_i$, for each non-leaf node $i$ are guided by routers, which are neural network functions linked to the generative model and defined as $\{r_{p,i} \mid i \in \mathbb{V}\backslash\mathbb{L}\}$. Here, $c_i = 0$ indicates the selection of the left child of the internal node $i$, aligning with the design proposed by Manduchi et al. (2023). As mentioned before, these decisions follow a Bernoulli distribution, $c_i \mid z_i \sim \text{Ber}\left(r_{p,i}\left(z_i\right)\right)$, which, in turn, induces the following path probability between a node $i$ and its parent node $pa(i)$:

$$p\left(c_{pa(i)\to i} \mid z_{pa(i)}\right) = \text{Ber}\left(c_{pa(i)\to i} \mid r_{p,pa(i)}(z_{pa(i)})\right). \tag{11}$$

Each decision made at a parent node determines the selection of the subsequent child node along the path. Consequently, the generative process progresses recursively until a leaf node $l$ is reached. Importantly, each decision path $\mathcal{P}_l$ from the root to the leaf $l$ is associated with a distinct set of decisions, which, in turn, determine the probabilities of traversing to different child nodes. These path probabilities, computed by the routers, play a critical role in shaping the distribution of the generated samples. The path $\mathcal{P}_l$ can be defined by the set of traversed nodes in the path. Moreover, let $z_{\mathcal{P}_l} = \{z_i \mid i \in \mathcal{P}_l\}$ denote the set of latent embeddings for each node in the path $\mathcal{P}_l$. Then, the prior probability of the latent embeddings and the path given the tree structure $\mathcal{T}$ corresponds to

$$p_\theta\left(z_{\mathcal{P}_l}, \mathcal{P}_l\right) = p\left(z_0\right) \prod_{i\in\mathcal{P}_l\backslash\{0\}} \underbrace{p\left(c_{pa(i)\to i} \mid z_{pa(i)}\right)}_{\text{decision probability}} \underbrace{p\left(z_i \mid z_{pa(i)}\right)}_{\text{sample probability}}. \tag{12}$$

To conclude the generative process, $x$ is obtained based on the latent embeddings of a selected leaf $l$. Nevertheless, assumptions on the distribution of the inputs are required. For real-valued $x$, such as in colored datasets, Manduchi et al. (2023) assume a Gaussian distribution. For grayscale images, they consider the Bernoulli distribution. Thus,

$$p_\theta\left(x \mid z_{\mathcal{P}_l}, \mathcal{P}_l\right) = \begin{cases} \mathcal{N}\left(x \mid \mu_{x,l}\left(z_l\right), \sigma_{x,l}^2\left(z_l\right)\right) & \text{for colored datasets,} \\ \text{Ber}\left(x \mid \mu_{x,l}\left(z_l\right)\right) & \text{for grayscale datasets,} \end{cases} \tag{13}$$

where $\{\mu_{x,l}, \sigma_{x,l} \mid l \in \mathbb{L}\}$ or $\{\mu_{x,l} \mid l \in \mathbb{L}\}$, respectively, are implemented as leaf-specific neural networks called "decoders". Typically, a simplification is made by assuming $\sigma_{x,l}^2(z_l) = I$ for convenience.

### INFERENCE MODEL

The inference model of TreeVAE, depicted on the left side of Figure 7, introduces a deterministic bottom-up pass to incorporate the conditioning on $x$, distinguishing it from the generative model. This bottom-up process is akin to the framework proposed by Sønderby et al. (2016). In this hierarchical structure, the bottom-up deterministic variables depend on each other via a series of neural network operations, specifically MLP of the same architecture as the transformation MLP defined previously,

$$\mathbf{d}_h = \text{MLP}\left(\mathbf{d}_{h+1}\right). \tag{14}$$

The first deterministic variable in this chain, denoted as $\mathbf{d}_H$, serves as the output of an encoder neural network. This encoder accepts the input image $x$ and transforms it into the flattened, low-dimensional vector $\mathbf{d}_H$. $H$ corresponds to both the number of deterministic variables involved in the bottom-up process and the maximum depth to which the tree structure can grow during training as further explained in Section A.2.

The tree structure is shared between the inference and the generative model, though adjustments are made to the node-specific parameterizations of the Gaussian distributions. Notably, in the inference phase, information is introduced through conditioning on $x$, influencing the distribution of latent embeddings across all nodes. Hence, similar to Sønderby et al. (2016), the means $\boldsymbol{\mu}_{q,i}$ and variances $\boldsymbol{\sigma}_{q,i}^2$ of the variational posterior distribution for each node $i$ are calculated using dense linear network layers conditioned on the deterministic variable of the same depth as the node $i$:

$$\hat{\boldsymbol{\mu}}_{q,i} = \text{Linear}\left(\mathbf{d}_{\text{depth}(i)}\right), \quad i \in \mathbb{V} \tag{15}$$

$$\hat{\boldsymbol{\sigma}}_{q,i}^2 = \text{Softplus}\left(\text{Linear}\left(\mathbf{d}_{\text{depth}(i)}\right)\right), \quad i \in \mathbb{V} \tag{16}$$

$$\boldsymbol{\sigma}_{q,i} = \frac{1}{\hat{\boldsymbol{\sigma}}_{q,i}^{-2} + \boldsymbol{\sigma}_{p,i}^{-2}}, \quad \boldsymbol{\mu}_{q,i} = \frac{\hat{\boldsymbol{\mu}}_{q,i}\hat{\boldsymbol{\sigma}}_{q,i}^{-2} + \boldsymbol{\mu}_{p,i}\boldsymbol{\sigma}_{p,i}^{-2}}{\hat{\boldsymbol{\sigma}}_{q,i}^{-2} + \boldsymbol{\sigma}_{p,i}^{-2}}, \tag{17}$$

where the subscript $q$ denotes the parameters specific to the inference model. Given these posterior parameters, the following equations determine the variational distributions of the latent embeddings for the root and the succeeding nodes in the tree structure:

$$q\left(\mathbf{z}_0 \mid \boldsymbol{x}\right) = \mathcal{N}\left(\mathbf{z}_0 \mid \mu_{q,0}(\boldsymbol{x}), \sigma_{q,0}^2(\boldsymbol{x})\right), \tag{18}$$

$$q_\phi\left(\mathbf{z}_i \mid \mathbf{z}_{pa(i)}\right) = \mathcal{N}\left(\mathbf{z}_i \mid \mu_{q,i}\left(\mathbf{z}_{pa(i)}\right), \sigma_{q,i}^2\left(\mathbf{z}_{pa(i)}\right)\right), \quad \forall i \in \mathcal{P}_l. \tag{19}$$

The routers in the inference model maintain the same architecture as those in the generative model. Nevertheless, in the inference process, decisions are now conditioned on $x$. This means that while the routers retain their original structure, the distribution of decision variables $c_i$ at node $i$ now depends on the deterministic variable of the corresponding depth, $c_i \mid \boldsymbol{x} \sim \text{Ber}(r_{q,i}(\mathbf{d}_{\text{depth}(i)}))$, resulting in the following variational path probability between a node $i$ and its parent node $pa(i)$:

$$q\left(c_{pa(i) \to i} \mid \boldsymbol{x}\right) = q\left(c_i \mid \mathbf{d}_{\text{depth}(pa(i))}\right) = \text{Ber}\left(c_{pa(i) \to i} \mid r_{q,pa(i)}\left(\mathbf{d}_{\text{depth}(pa(i))}\right)\right), \tag{20}$$

Finally, given the variational distributions of the latent embeddings equation 18 & equation 19 and the variational distributions of the decisions equation 20, Manduchi et al. (2023) construct the variational posterior distribution of the latent embeddings and paths:

$$q\left(\mathbf{z}_{\mathcal{P}_l}, \mathcal{P}_l \mid \boldsymbol{x}\right) = q\left(\mathbf{z}_0 \mid \boldsymbol{x}\right) \prod_{i \in \mathcal{P}_l \setminus \{0\}} q\left(c_{pa(i) \to i} \mid \boldsymbol{x}\right) q\left(\mathbf{z}_i \mid \mathbf{z}_{pa(i)}\right). \tag{21}$$

## A.2 MODEL TRAINING

TreeVAE entails the training of various components, involving both the parameters of the neural network layers present in the inference and generative models, as detailed in Section A.1, and the binary tree structure $\mathcal{T}$. The training process alternates between optimizing the model parameters with a fixed tree structure and expanding the tree.

During a training iteration given the current tree structure, the neural layer parameters are optimized. These include the parameters of the inference model and the generative model, encompassing the encoder ($\mu_{q,0}, \sigma_{q,0}$), the bottom-up MLPs from equation 14, the dense linear layers from equation 15 & equation 16, the transformations ($\{(\mu_{p,i}, \sigma_{p,i}), (\mu_{q,i}, \sigma_{q,i}) \mid i \in \mathbb{V} \setminus \{0\}\}$), the routers ($\{r_{p,i}, r_{q,i} \mid i \in \mathbb{V} \setminus \mathbb{L}\}$), and the decoders ($\{\mu_{x,l}, \sigma_{x,l} \mid l \in \mathbb{L}\}$). The objective in training these parameters is to maximize the likelihood of the data given the learned tree structure $\mathcal{T}$, denoted as $p(\boldsymbol{x} \mid \mathcal{T})$. This corresponds to modeling the distribution of real data via the hierarchical latent embeddings. To compute $p(\boldsymbol{x} \mid \mathcal{T})$, the latent embeddings must be marginalized out from the joint distribution:

$$p(\boldsymbol{x} \mid \mathcal{T}) = \sum_{l \in \mathbb{L}} \int_{\boldsymbol{z}_{\mathcal{P}_l}} p(\boldsymbol{x}, \boldsymbol{z}_{\mathcal{P}_l}, \mathcal{P}_l) = \sum_{l \in \mathbb{L}} \int_{\boldsymbol{z}_{\mathcal{P}_l}} p_\theta(\boldsymbol{z}_{\mathcal{P}_l}, \mathcal{P}_l) \, p_\theta(\boldsymbol{x} \mid \boldsymbol{z}_{\mathcal{P}_l}, \mathcal{P}_l). \qquad (22)$$

Similar to other Variational Autoencoders (Kingma & Welling, 2014; Rezende et al., 2014), the optimization of TreeVAE ultimately comes down to maximizing the ELBO. As shown by Manduchi et al. (2023), the ELBO in this setting can be written as follows:

$$\mathcal{L}(\boldsymbol{x} \mid \mathcal{T}) := \underbrace{\mathbb{E}_{q(\boldsymbol{z}_{\mathcal{P}_l}, \mathcal{P}_l \mid \boldsymbol{x})}[\log p(\boldsymbol{x} \mid \boldsymbol{z}_{\mathcal{P}_l}, \mathcal{P}_l)]}_{\mathcal{L}_{rec}} - \underbrace{\mathrm{KL}(q(\boldsymbol{z}_{\mathcal{P}_l}, \mathcal{P}_l \mid \boldsymbol{x}) \| p(\boldsymbol{z}_{\mathcal{P}_l}, \mathcal{P}_l))}_{\mathrm{KL}_{root} + \mathrm{KL}_{nodes} + \mathrm{KL}_{decisions}}. \qquad (23)$$

Hereby, we distinguish between the reconstruction term and the KL divergence term between the variational posterior and the prior of the tree, which can be further decomposed into contributions from the root, the remaining nodes, and decisions, as indicated in Equation equation 23. Both terms are approximated using Monte Carlo (MC) sampling during training, as elaborated further in Manduchi et al. (2023).

To grow the binary tree structure $\mathcal{T}$, TreeVAE begins with a simple tree configuration, typically composed of a root and two leaves. This initial structure is trained for a defined number of epochs, optimizing the ELBO. Subsequently, the model iteratively expands the tree by attaching two new child nodes to a current leaf node in the model. In their approach, Manduchi et al. (2023) opted to expand the tree by selecting nodes with the highest number of assigned samples, thus implicitly encouraging balanced leaves. The sub-tree formed by the new leaves and the parent node undergoes training for another number of epochs, keeping the weights of the remaining model frozen. This expansion process continues until either the tree reaches its maximum depth $H$, a predefined maximum number of effective leaves, or another predefined condition is met. Optionally, after the tree has been expanded, all parameters in the model may be fine-tuned for another predefined number of epochs. Finally, the tree is pruned to remove empty branches.

To improve the clustering performance, especially for colored images, Manduchi et al. (2023) enhance TreeVAE with contrastive learning. This addition enables the model to encode prior knowledge on data invariances through augmentations, facilitating the learning process and better capturing meaningful relationships within complex data. By incorporating contrastive objectives into the training process, TreeVAE becomes more adept at retrieving semantically meaningful clusters from colored image data.

# B  DIFFUSION MODELS

Diffusion models have garnered considerable attention in recent years for their remarkable image-generation capabilities, outperforming traditional GANs in achieving high-quality results. This surge in interest has led to the development of numerous diffusion-based models, with the initial concept being introduced by Sohl-Dickstein et al. (2015), drawing inspiration from principles rooted in thermodynamics. In this section, we delve into one of the first and most well-known diffusion models, namely the DDPM, outlined in Section B.1. Additionally, we explore one possible integration of DDPM with VAEs to leverage the interpretable latent space offered by VAEs along with the superior generation quality of DDPM. Note that we use the DDIM instantiation at inference time.

## B.1  DIFFUSION DENOISING PROBABILISTIC MODELS

Diffusion Denoising Probabilistic Models (DDPM), introduced by Ho et al. (2020), are latent variable models that consist of two opposed processes. The main idea behind DDPM involves iteratively

adding noise to the original image, thereby progressively degrading the signal with each step until only noise remains. In a second step, the image is successively reconstructed through a denoising process, employing a learned function to remove the noise. Therefore, DDPM essentially entail a forward noising process followed by a reverse denoising process, aiming to restore the original image from its noisy counterpart. Subsequently, both these processes will be explored in more detail.

### FORWARD PROCESS

The forward process, also known as the diffusion process or forward noising process, serves a similar purpose as the inference model in VAEs. It operates as a Markov chain that gradually introduces noise to the data signal $\boldsymbol{x}_0$ over $T$ steps, where the intermediate states of the deformed data are denoted as $\boldsymbol{x}_t$ for $t = 1, \ldots, T$. This process follows a trajectory determined by a noise schedule $\beta_1, \ldots, \beta_T$, which controls the rate at which the original data is degraded. While it is possible to learn the variances $\beta_t$ via reparametrization, they are often chosen as hyperparameters with a predetermined schedule (Ho et al., 2020). Assuming Gaussian noise, the forward process can be represented as follows:

$$q\left(\boldsymbol{x}_{1:T} \mid \boldsymbol{x}_0\right) = \prod_{t=1}^{T} q\left(\boldsymbol{x}_t \mid \boldsymbol{x}_{t-1}\right) \tag{24}$$

$$q\left(\boldsymbol{x}_t \mid \boldsymbol{x}_{t-1}\right) = \mathcal{N}\left(\sqrt{1 - \beta_t}\boldsymbol{x}_{t-1}, \beta_t \boldsymbol{I}\right) \tag{25}$$

Importantly, $\boldsymbol{x}_t$ for any step $t$ can be sampled directly given $\boldsymbol{x}_0$ using:

$$q\left(\boldsymbol{x}_t \mid \boldsymbol{x}_0\right) = \mathcal{N}\left(\sqrt{\bar{\alpha}_t}\boldsymbol{x}_0, \left(1 - \bar{\alpha}_t\right)\boldsymbol{I}\right), \tag{26}$$

where $\alpha_t = (1 - \beta_t)$ and $\bar{\alpha}_t = \prod_{s=1}^{t} \alpha_s$. This property significantly boosts training efficiency by eliminating the need to sample every state between $\boldsymbol{x}_0$ and $\boldsymbol{x}_t$. Additionally, when conditioned on $\boldsymbol{x}_0$, the posteriors of the forward process are tractable and can be determined in closed form:

$$q\left(\boldsymbol{x}_{t-1} \mid \boldsymbol{x}_t, \boldsymbol{x}_0\right) = \mathcal{N}\left(\boldsymbol{x}_{t-1}; \tilde{\boldsymbol{\mu}}_t\left(\boldsymbol{x}_t, \boldsymbol{x}_0\right), \tilde{\beta}_t \boldsymbol{I}\right), \tag{27}$$

$$\text{where} \quad \tilde{\boldsymbol{\mu}}_t\left(\boldsymbol{x}_t, \boldsymbol{x}_0\right) = \frac{\sqrt{\bar{\alpha}_{t-1}}\beta_t}{1 - \bar{\alpha}_t}\boldsymbol{x}_0 + \frac{\sqrt{\alpha_t}\left(1 - \bar{\alpha}_{t-1}\right)}{1 - \bar{\alpha}_t}\boldsymbol{x}_t \quad \text{and} \quad \tilde{\beta}_t = \frac{1 - \bar{\alpha}_{t-1}}{1 - \bar{\alpha}_t}\beta_t. \tag{28}$$

### REVERSE PROCESS

The reverse process, also known as the reverse denoising process or backward process, is comparable to the generative model in Variational Autoencoders. Beginning with pure Gaussian noise, the reverse process governs the sample generation by progressively eliminating noise through a sequence of $T$ denoising steps. These steps are facilitated by time-dependent, learnable Gaussian transitions that adhere to the Markov property, meaning each transition relies solely on the state of the previous time step in the Markov chain. This denoising sequence reverses the noise addition steps of the forward process, enabling the gradual recovery of the original data.

In summary, the reverse process models a complex target data distribution by sequentially transforming simple distributions through a generative Markov chain. This approach overcomes the traditional tradeoff between tractability and flexibility in probabilistic models (Sohl-Dickstein et al., 2015), leading to a flexible and efficient generative model:

$$p\left(\boldsymbol{x}_{0:T}\right) = p\left(\boldsymbol{x}_T\right) \prod_{t=1}^{T} p_\theta\left(\boldsymbol{x}_{t-1} \mid \boldsymbol{x}_t\right) \tag{29}$$

$$p_\theta\left(\boldsymbol{x}_{t-1} \mid \boldsymbol{x}_t\right) = \mathcal{N}\left(\boldsymbol{\mu}_\theta\left(\boldsymbol{x}_t, t\right), \boldsymbol{\Sigma}_\theta\left(\boldsymbol{x}_t, t\right)\right) \tag{30}$$

Ho et al. (2020) keep the time dependent variances $\boldsymbol{\Sigma}_\theta\left(\boldsymbol{x}_t, t\right) = \sigma_t^2 \boldsymbol{I}$ constant, typically choosing $\sigma_t^2 = \beta_t$ or $\sigma_t^2 = \tilde{\beta}_t = \frac{1 - \bar{\alpha}_{t-1}}{1 - \bar{\alpha}_t}\beta_t$. Therefore, only the time-dependent posterior mean function $\boldsymbol{\mu}_\theta$ is learned during training. In fact, Ho et al. (2020) further suggest the following reparametrization of the posterior means:

$$\boldsymbol{\mu}_\theta\left(\boldsymbol{x}_t, t\right) = \tilde{\boldsymbol{\mu}}_t\left(\boldsymbol{x}_t, \frac{1}{\sqrt{\bar{\alpha}_t}}\left(\boldsymbol{x}_t - \sqrt{1 - \bar{\alpha}_t}\boldsymbol{\epsilon}_\theta\left(\boldsymbol{x}_t\right)\right)\right) = \frac{1}{\sqrt{\alpha_t}}\left(\boldsymbol{x}_t - \frac{\beta_t}{\sqrt{1 - \bar{\alpha}_t}}\boldsymbol{\epsilon}_\theta\left(\boldsymbol{x}_t, t\right)\right), \tag{31}$$

where $\boldsymbol{\epsilon}_\theta$ is a learnable function that predicts the noise at any given time step $t$.

MODEL TRAINING

Training a diffusion model corresponds to optimizing the reverse Markov transitions to maximize the likelihood of the data. This is achieved by maximizing the ELBO or, equivalently, minimizing the variational upper bound on the negative log-likelihood:

$$\mathcal{L} = \mathbb{E}_q[\underbrace{\text{KL}\left(q\left(\boldsymbol{x}_T \mid \boldsymbol{x}_0\right) \| p\left(\boldsymbol{x}_T\right)\right)}_{\mathcal{L}_T} + \sum_{t>1} \underbrace{\text{KL}\left(q\left(\boldsymbol{x}_{t-1} \mid \boldsymbol{x}_t, \boldsymbol{x}_0\right) \| p_\theta\left(\boldsymbol{x}_{t-1} \mid \boldsymbol{x}_t\right)\right)}_{\mathcal{L}_{t-1}} - \underbrace{\log p_\theta\left(\boldsymbol{x}_0 \mid \boldsymbol{x}_1\right)}_{\mathcal{L}_0}]$$

(32)

For small diffusion rates $\beta_t$, the forward and the reverse processes share the same functional form, following Gaussian distributions (Sohl-Dickstein et al., 2015). Thus, the KL divergence terms between the posteriors of the forward process equation 27 and the reverse process equation 29 have a closed form. Ho et al. (2020) further simplified the training process by introducing a more efficient training objective. Given the assumption of fixed variances, $\mathcal{L}_{t-1}$ can be written as:

$$\mathcal{L}_{t-1} = \mathbb{E}_q\left[\frac{1}{2\sigma_t^2}\left\|\tilde{\boldsymbol{\mu}}_t\left(\boldsymbol{x}_t, \boldsymbol{x}_0\right) - \boldsymbol{\mu}_\theta\left(\boldsymbol{x}_t, t\right)\right\|^2\right] + C,$$

(33)

where $C$ is a constant that does not depend on the model parameters $\theta$. Thus, the reverse process mean function, $\boldsymbol{\mu}_\theta$, is optimized to predict $\tilde{\boldsymbol{\mu}}_t$, the fixed noisy mean function at time step $t$ from the forward process equation 28. However, instead of directly comparing $\boldsymbol{\mu}_\theta$ and $\tilde{\boldsymbol{\mu}}_t$, by reparametrization, the model can be trained to predict the noise $\boldsymbol{\epsilon}$ at any given time step $t$ which results in more stable training results according to Ho et al. (2020). Thus, equation 33 can be further rewritten as

$$\mathcal{L}_{t-1} - C = \mathbb{E}_{\boldsymbol{x}_0, \boldsymbol{\epsilon}}\left[\frac{\beta_t^2}{2\sigma_t^2 \alpha_t\left(1 - \bar{\alpha}_t\right)}\left\|\boldsymbol{\epsilon} - \boldsymbol{\epsilon}_\theta\left(\sqrt{\bar{\alpha}_t}\boldsymbol{x}_0 + \sqrt{1 - \bar{\alpha}_t}\boldsymbol{\epsilon}, t\right)\right\|^2\right],$$

(34)

where $\boldsymbol{\epsilon}_\theta$ is a function parametrized as a neural network that maintains equal dimensionality for input and output. The preferred architecture for $\boldsymbol{\epsilon}_\theta$ is a U-Net (Ronneberger et al., 2015), as used by Ho et al. (2020), Dhariwal & Nichol (2021), and Pandey et al. (2022). The training of $\boldsymbol{\epsilon}_\theta$ involves multiple epochs, where for each sample $\boldsymbol{x}_0$ of the original data, a time step $t$ within the diffusion sequence $1, \ldots, T$ is chosen at random. Subsequently, some noise $\boldsymbol{\epsilon} \sim \mathcal{N}(\boldsymbol{0}, \boldsymbol{I})$ is sampled for that specific time step, which needs to be predicted by $\boldsymbol{\epsilon}_\theta$. To optimize the model, a gradient step is computed with respect to the loss function detailed in Equation equation 34. Therefore, this training procedure does not require iterating through every step of the diffusion model, circumventing the issue of slow sample generation that is typical for diffusion models.

# C  ADDITIONAL QUALITATIVE RESULTS

## C.1  GENERATIONS ON MNIST

Figure 8 presents an additional plot similar to those in Figure 4 and Figure 5 from the main text. This plot illustrates the generated images of the TreeDiffusion model when trained on the MNIST dataset. In each of these plots, we display randomly generated images for each cluster. Below each set of leaf-specific images, we provide a normalized histogram showing the distribution of predicted classes by an independent ResNet-50 classifier that has been pre-trained on the training data of the respective dataset. This visualization helps in understanding how well the model can generate distinct and meaningful clusters in the context of different datasets.

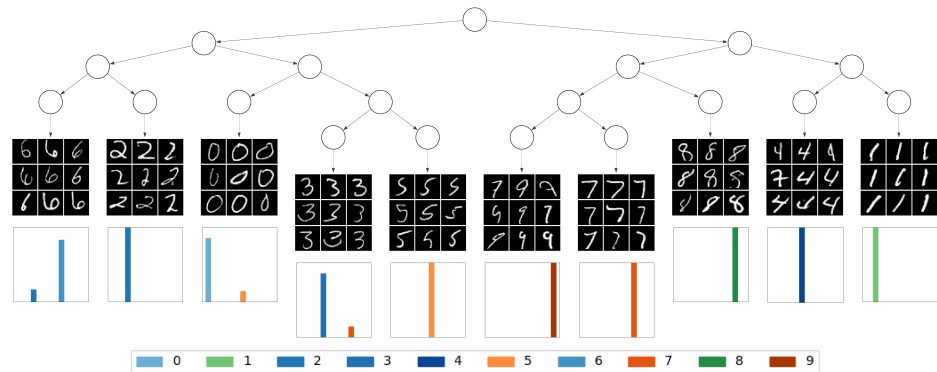

Figure 8: TreeDiffusion model trained on MNIST. For each cluster, random newly generated images are displayed. Below each set of images, a normalized histogram (ranging from 0 to 1) shows the distribution of predicted classes from an independent, pre-trained classifier on MNIST for all newly generated images in each leaf with a significant probability of reaching that leaf.

## C.2    ADDITIONAL GENERATION EXAMPLES FOR TREEVAE VS. TREEDIFFUSION

L0: p=0.00 L1: p=0.00 L2: p=0.00 L3: p=0.00 L4: p=0.01 L5: p=0.00 L6: p=0.00 L7: p=0.81 L8: p=0.00 L9: p=0.18

TreeVAE

TreeDiffusion

L0: p=0.00 L1: p=0.75 L2: p=0.00 L3: p=0.00 L4: p=0.00 L5: p=0.25 L6: p=0.00 L7: p=0.00 L8: p=0.00 L9: p=0.00

TreeVAE

TreeDiffusion

L0: p=0.01 L1: p=0.05 L2: p=0.00 L3: p=0.00 L4: p=0.94 L5: p=0.00 L6: p=0.00 L7: p=0.00 L8: p=0.00 L9: p=0.00

TreeVAE

TreeDiffusion

L0: p=0.00 L1: p=0.00 L2: p=0.00 L3: p=0.03 L4: p=0.04 L5: p=0.00 L6: p=0.00 L7: p=0.94 L8: p=0.00 L9: p=0.00

TreeVAE

TreeDiffusion

L0: p=0.00 L1: p=0.00 L2: p=0.00 L3: p=0.00 L4: p=0.02 L5: p=0.00 L6: p=0.00 L7: p=0.00 L8: p=0.00 L9: p=0.98

TreeVAE

TreeDiffusion

L0: p=0.00 L1: p=0.00 L2: p=0.00 L3: p=0.84 L4: p=0.00 L5: p=0.00 L6: p=0.15 L7: p=0.00 L8: p=0.01 L9: p=0.00

TreeVAE

TreeDiffusion

Figure 9: For each example, we show image generations from every leaf of the (top) TreeVAE and (bottom) TreeDiffusion model, both trained on the MNIST dataset. Each row shows the generated images from all leaves of the respective model, starting with the same root sample. The corresponding leaf probabilities are shown above each image and are identical across both models by design.

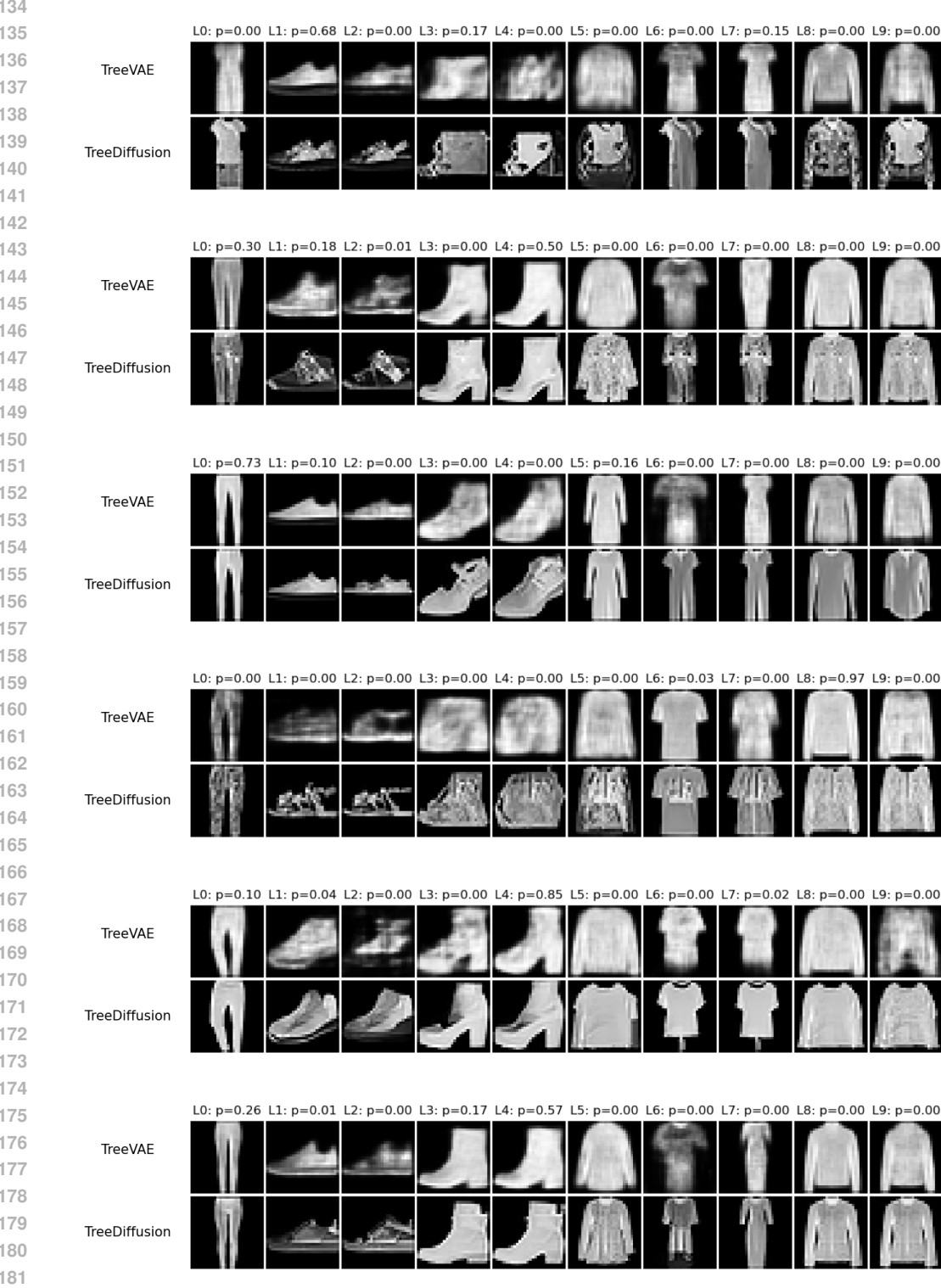

Figure 10: For each example, we show image generations from every leaf of the (top) TreeVAE and (bottom) TreeDiffusion model, both trained on the FashionMNIST dataset. Each row shows the generated images from all leaves of the respective model, starting with the same root sample. The corresponding leaf probabilities are shown above each image and are identical across both models by design.

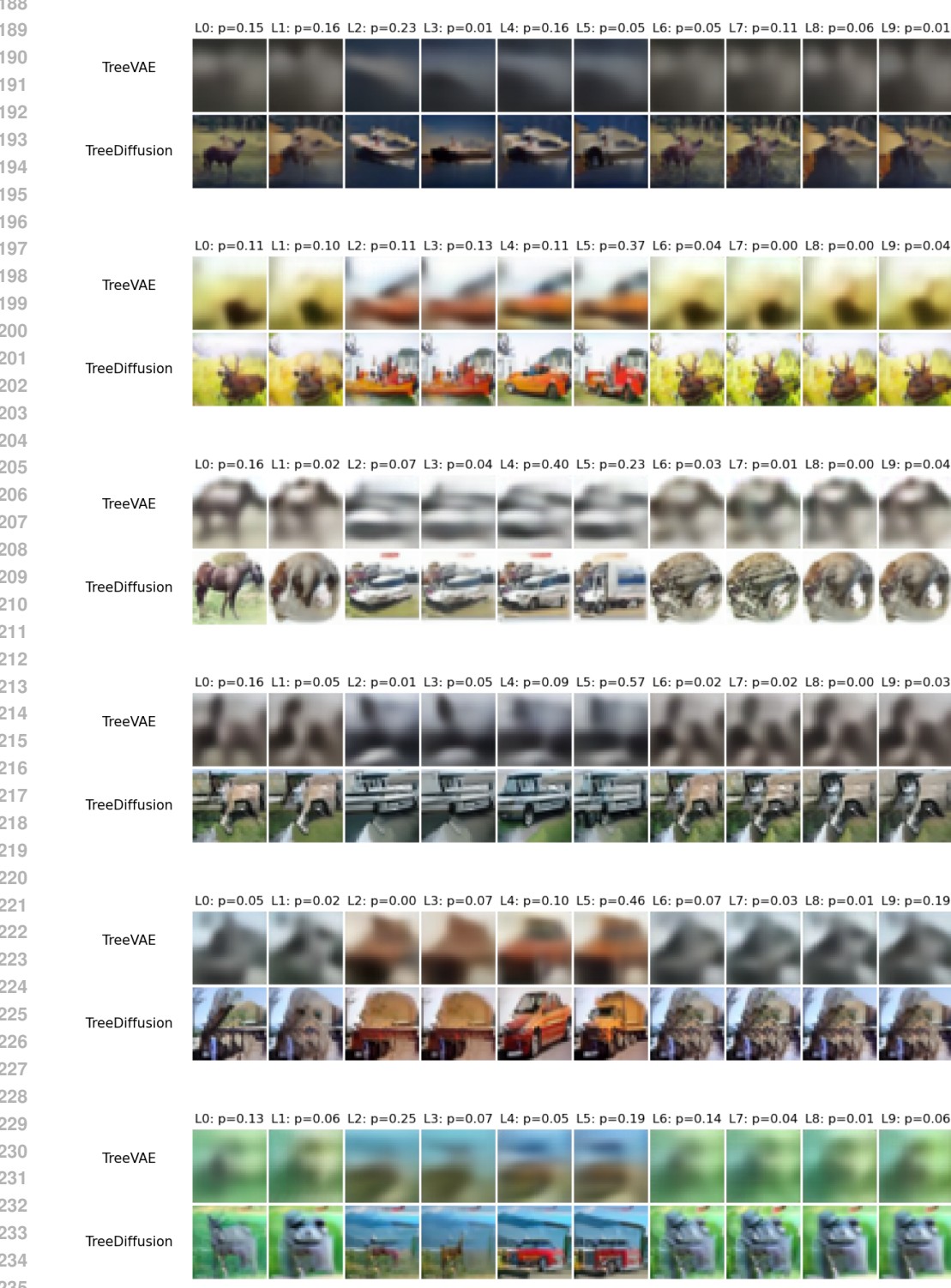

Figure 11: For each example, we show image generations from every leaf of the (top) TreeVAE and (bottom) TreeDiffusion model, both trained on the CIFAR-10 dataset. Each row shows the generated images from all leaves of the respective model, starting with the same root sample. The corresponding leaf probabilities are shown above each image and are identical across both models by design.

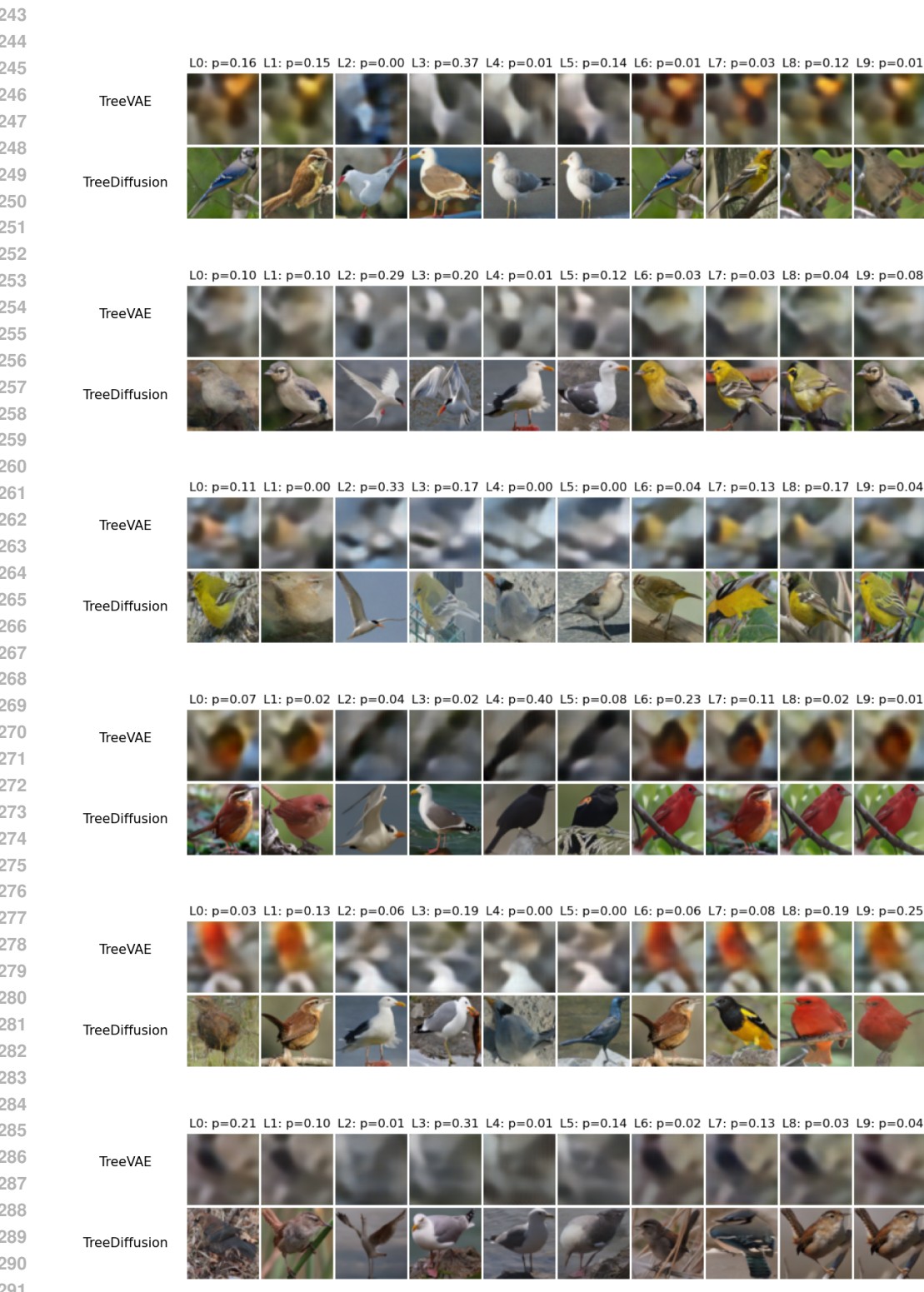

Figure 12: For each example, we show image generations from every leaf of the (top) TreeVAE and (bottom) TreeDiffusion model, both trained on the CUBICC dataset. Each row shows the generated images from all leaves of the respective model, starting with the same root sample. The corresponding leaf probabilities are shown above each image and are identical across both models by design.

# D IMPLEMENTATION DETAILS

TREEVAE TRAINING

Table 4: Overview of training configurations for TreeVAE, including model parameters, training hyperparameters, and contrastive learning specifics across datasets.

| Data resolution | 28x28x1 | 32x32x3 | 64x64x3 |
|---|---|---|---|
| Encoder/Decoder Types | cnn1 | cnn1 | cnn2 |
| Max. tree depth | 7 | 7 | 7 |
| Max. clusters | 10 | 10 | 10 |
| Representation dimensions | 4 | 4 | 4 |
| # of latent channels | 16 | 64 | 64 |
| # of bottom-up channels | 32 | 128 | 128 |
| Grow | True | True | True |
| Prune | True | True | True |
| Activation of last layer | sigmoid | mse | mse |
| Optimizer | Adam(lr=1e-3) | Adam(lr=1e-3) | Adam(lr=1e-3) |
| Effective batch size | 256 | 256 | 128 |
| # of initial epochs | 150 | 150 | 150 |
| # of smalltree epochs | 150 | 150 | 150 |
| # of intermediate epochs | 80 | 0 | 0 |
| # of fine-tuning epochs | 200 | 0 | 0 |
| lr decay rate | 0.1 | 0.1 | 0.1 |
| lr decay step size | 100 | 100 | 100 |
| Weight decay | 1e-5 | 1e-5 | 1e-5 |
| Contrastive augmentations | False | True | True |
| Augmentation method | None | InfoNCE | InfoNCE |
| Augmentation weight | None | 100 | 100 |

We utilize a modified variant of the TreeVAE model, employing CNNs for its operations. The representations within TreeVAE are maintained in a 3-dimensional format, where the first dimension signifies the number of channels, while the subsequent two dimensions denote the spatial dimensions of the representations. Consequently, the deterministic variables in the bottom-up pathway possess a dimensionality of (# of bottom-up channels, representation dimension, representation dimension), while the stochastic variables in the top-down tree structure have a dimensionality of (# of latent channels, representation dimension, representation dimension). Table 4 provides details on the remaining parameters utilized for the TreeVAE model and its training. For more information, please refer to the code.

TREEDIFFUSION TRAINING

Table 5 provides details on the remaining parameters utilized for the diffusion model and its training. For more information, please refer to the code.

Table 5: Overview of training configurations for the DDPM of the TreeDiffusion, including model parameters and training hyperparameters.

| Data resolution | 28x28x1 | 32x32x3 | 64x64x3 |
|---|---|---|---|
| Noise Schedule | Linear(1e-4, 0.02) | Linear(1e-4, 0.02) | Linear(1e-4, 0.02) |
| # of U-Net channels | 64 | 128 | 128 |
| Scale(s) of attention block | [16] | [16,8] | [16,8] |
| # of res. blocks per scale | 2 | 2 | 2 |
| Channel multipliers | (1,2,2,2) | (1,2,2,2) | (1,2,2,2,4) |
| Dropout | 0.3 | 0.3 | 0.3 |
| Diffusion loss type | L2 | L2 | L2 |
| Optimizer | Adam(lr=2e-4) | Adam(lr=2e-4) | Adam(lr=2e-4) |
| Effective batch size | 256 | 256 | 32 |
| # of lr annealing steps | 5000 | 5000 | 5000 |
| Grad. clip threshold | 1.0 | 1.0 | 1.0 |
| EMA decay rate | 0.9999 | 0.9999 | 0.9999 |

