# OpenReview forum: "Hierarchical Clustering for Conditional Diffusion in Image Generation"
_ICLR.cc/2025/Conference — Submitted to ICLR 2025_

### Official Review · Reviewer_E4q2 · 2024-11-01

**Soundness:** 3
**Presentation:** 3
**Contribution:** 2
**Rating:** 5
**Confidence:** 4

**Summary:**

The paper introduces TreeDiffusion, a two-stage framework that combines a hierarchical clustering model (based on TreeVAE (Manduchi et al., 2023)) with a diffusion model to improve the generation quality. This approach addresses limitations in previous TreeVAE, where the clustering models could achieve hierarchical clustering, but suffer from poor generations. TreeDiffusion first learns a hierarchical latent tree and parameterize a diffusion model conditioned on the raw prediction sampled from the latent.

**Strengths:**

- The paper extends the idea of TreeVAE to TreeDiffusion, leveraging a diffusion model to improve the low-quality generation issue exposed in TreeVAE. With such a design, the model can conceptually provide both hierarchical representations and conditional generations conditioned on these representations.

- The idea of the paper is clear and straightforward to follow. The paper is overall clear writtent, facilitating a smooth reading experience.

- The experiments included in the paper strongly support one of the authors' claim, where the generation quality is significantly improved in terms of both FiD measure and visualized images.

**Weaknesses:**

- There are some defaults in the math presentations of the paper. For example, it is not clear why in Eq(1) and Eq(2) the LHS equals to the RHS.

- The method part does not explain how the model is trained. The authors only simply point out the path latents $z_{P_l}$ are used as conditions of the denoising UNet. It could be unclear to the readers who are not familiar with diffusion models. Although in appendix B the authors provide the parameterization of diffusion models, most of the depiction is from the original DDPM and DDIM paper. The authors should give credits to the previous paper and use more space in explaining the proposed method.


- The paper does not conduct sufficient analysis on why the incorporation of diffusion models could improve TreeVAE and DiffuseVAE. Mathematically, does the TreeDiffusion objective make the ELBO shown in Eq (3) tightened? Does TreeDiffusion makes the latents and the images has better alignment in distribution or it improves the log-likelihood part? Alternatively, does the improvement comes from the increasing of the total parameter of the model (as we include Unet as additional parameters)?

- Although the authors claim TreeDiffusion demonstrates a new conditional generation control, I did not find clear evidences on how the the latent conditions provide a better or novel control in the generation. More specifically, does the hierarchical latents provide instance-level or any fine-grained conditioning in the generation?

- Considering the previous point, the proposed method seems to simply uses a diffusion model as a post-processing way to extend the TreeVAE and DiffuseVAE only in generation quality, while it could be more interesting if the authors could examine whether there are more outcomes in the hybrid model.

**Questions:**

Please see the weakness.

---

> ### Author Response · Authors · 2024-11-24
>
> We thank the reviewer for their thoughtful comments. Below, we address the identified weaknesses and questions in the same order as listed by the reviewer.
>
> • This follows directly from the TreeVAE paper. To clarify, we provide a more detailed explanation of these equations in the appendix, as presented by the original TreeVAE authors.
>
> • The model training process will be clarified in the revised paper. Specifically, we will provide a more detailed explanation of how the hierarchical latent path information from TreeVAE is used as a conditioning signal in the denoising UNet. Regarding credit, we cite the DDPM and DDIM papers multiple times. However, we will ensure clearer acknowledgment of their contributions and elaborate further on how our approach extends these methods.
>
> • TreeDiffusion improves the ELBO compared to both models. The TreeDiffusion ELBO is a modified version of the DiffuseVAE ELBO, as described in Eq. (10) of their paper. The key differences are that the conditioning signal $y$ is not fixed but treated as a latent representation $z$ due to the unsupervised setting, and the VAE ELBO is replaced with the TreeVAE ELBO. As a result, the TreeDiffusion ELBO consists of a term for the TreeVAE ELBO and an additional term for the diffusion model.
>
> This design allows TreeDiffusion to improve upon TreeVAE regarding the ELBO by incorporating the U-Net for the diffusion model. Furthermore, since TreeVAE has a tighter ELBO than a standard VAE (as shown in Table 2 of the TreeVAE paper), TreeDiffusion also improves upon the DiffuseVAE in terms of ELBO.
>
> • The hierarchical latents offer cluster-level conditioning based on the learned hierarchical clusters. We did not explore more fine-grained conditioning in this work, as we focus on leveraging the hierarchical structure to improve cluster-based image generation.
>
> • We assume that by "hybrid model," you are referring to a model trained end-to-end. In our experiments, we observed that training a fully end-to-end hybrid model led to a deterioration in clustering performance over the course of training. Specifically, the model tended to collapse to a single path in the tree structure, effectively eliminating the clustering capabilities.

---

> > ### Comment · Reviewer_E4q2 · 2024-11-25
> >
> > Thank you for your detailed response. After reviewing the rebuttal, I find that some of my concerns remain unresolved. I would like to address the following questions for further clarification:
> >
> > 1. TreeDiffusion still refines the output of decoded leaf nodes, conditioned on the hierarchical latents produced by TreeVAE. This approach seems to combine the previous studies of TreeVAE and DiffuseVAE. However, what is the core conclusion derived from this design? Specifically, does TreeVAE demonstrate superior effectiveness in producing meaningful clusters or it only improves the decoding quality? This raises the question of why the TreeVAE is particularly suited for clustering, and do the improvements come from the power of the diffusion or the TreeVAE-cluster conditions in this context.
> >
> > 2. The rebuttal suggests that TreeVAE-based conditioning improves the performance of the diffusion model. Why does TreeVAE enhance the diffusion model’s performance, and why the alternative methods less effective? A comparative analysis could strengthen the understanding of TreeVAE’s unique utility in this framework.
> >
> > 3. Has the evidence demonstrated that the ELBO is tightened compared to standalone TreeVAE and DiffuseVAE? A more explicit discussion or proof in this regard would help clarify this point.
> >
> > Overall, I find the benefits of the proposed idea still unclear. I will keep my rating until these points are addressed.

---

### Official Review · Reviewer_dZcu · 2024-11-02

**Soundness:** 2
**Presentation:** 3
**Contribution:** 1
**Rating:** 3
**Confidence:** 4

**Summary:**

The paper introduces TreeDiffusion, a framework that enhances the image generation quality of hierarchical clustering models by conditioning a diffusion model on hierarchical features learned through TreeVAE. The approach leverages TreeVAE for generating latent hierarchical structures and subsequently applies a diffusion model for cluster-specific image generation. This two-stage setup aims to address the generative limitations of clustering-based VAEs, producing sharper and more diverse images within each cluster.

While the integration of hierarchical clustering with diffusion models is an interesting approach, the overall novelty of this paper is limited. The conditioning of diffusion models on latent structures, specifically from VAEs, is well-explored in generative modeling, as are hierarchical VAE approaches. Furthermore, the contribution of this work is ambiguous given that it primarily extends existing techniques without providing significant innovations in method or theory.

**Strengths:**

1. The paper presents a practical attempt to address clustering-based VAE’s limitations using diffusion conditioning, which enhances image quality and diversity in the generated samples.
2. The paper demonstrates the generative advantages of TreeDiffusion over TreeVAE, showing clearer images with reduced blurriness across datasets.

**Weaknesses:**

1. Both conditional diffusion models and hierarchical VAEs are well-established methods in generative modeling. This paper’s approach seems to be rather a simple combination of TreeVAE [4] and DiffusionVAE [2].
2. No quantitative comparison between the proposed approach and DiffusionVAE is provided, making it difficult to validate the effectiveness of the proposed TreeDiffusion.
3. The experimental section includes visualizations of cluster-specific generation results. However, as described in Section 3.1, TreeVAE is used with only minor adaptations, resulting in identical clustering performance between TreeVAE and TreeDiffusion. Therefore, these visualizations alone do not demonstrate the contribution of this paper and seem redundant.
4. The experiments are limited to comparison with TreeVAE, which does not offer a comprehensive evaluation of TreeDiffusion's performance. Comparisons with state-of-the-art methods, e.g. LDMs [1], DiffuseVAE [2], KNN-Diffusion [3] et al., would be necessary to validate TreeDiffusion’s superiority and clarify its practical benefits over established baselines.

[1] Rombach, Robin, et al. "High-resolution image synthesis with latent diffusion models." Proceedings of the IEEE/CVF conference on computer vision and pattern recognition. 2022.

[2] Pandey, Kushagra, et al. "DiffuseVAE: Efficient, Controllable and High-Fidelity Generation from Low-Dimensional Latents." Transactions on Machine Learning Research.

[3] Sheynin, Shelly, et al. "kNN-Diffusion: Image Generation via Large-Scale Retrieval." The Eleventh International Conference on Learning Representations.

[4] Manduchi, Laura, et al. "Tree variational autoencoders." Advances in Neural Information Processing Systems 36 (2023): 54952-54986.

**Questions:**

1. Diffusion models are known for their slow inference speeds. How do the inference speed and parameter overhead of TreeDiffusion compare to those of TreeVAE, e.g., the inference time and memory cost for generating one sample?
2. How can we generate a sample using TreeDiffusion during inference? If the second-stage diffusion model is conditioned on latent features provided by the TreeVAE encoder, does generating a sample require a reference image to extract these latent features?
3. The visual quality of DiffuseVAE appears to fall short of the results shown in the original paper. Could the authors clarify the cause of this difference? In the original DiffuseVAE, generated images seem to be much sharper and more diverse.
4. Class labels alone could provide useful conditioning signals for training diffusion models. Have the authors compared the performance of class conditioning with that of conditions derived from TreeVAE?

---

> ### Author Response · Authors · 2024-11-24
>
> We thank the reviewer for their thoughtful comments. Below, we address the identified weaknesses and questions.
>
> Regarding the remarks under Weaknesses:
>
> 1. As explained in more detail in our General Response, our approach goes beyond a simple combination of TreeVAE and DiffuseVAE. The key difference lies in conditioning the diffusion model on the hierarchical latent information from TreeVAE without refining its outputs. Unlike DiffuseVAE, which does not condition on the latent representations, our model relies exclusively on the latent hierarchical representations. We will update the paper to clarify this distinction.
> 2. We appreciate your suggestion. For the final version, we will include a comparison with DiffuseVAE in terms of FID, using a vanilla VAE as described in the DiffuseVAE paper. Additionally, we already provide metrics for a model that refines only the reconstructed outputs from TreeVAE during the reverse process, without conditioning on latent information, which aligns closely with DiffuseVAE.
> 3. Due to the time-intensive nature of training these models, it was not feasible to include additional baselines during the rebuttal period but we will consider this for the final version of the paper.
> 4. While TreeDiffusion and TreeVAE achieve the same clustering performance, TreeDiffusion significantly improves image quality. Furthermore, while TreeVAE performs well in clustering, its generated and reconstructed images often fail to distinctly represent different clusters, especially for colored datasets where generations were particularly poor. These visualizations aim to highlight not only the higher-quality images but also the distinctiveness and recognizability of the clusters, demonstrating that our model produces images that adhere more closely to the latent tree's clustering.
>
> Regarding the Questions:
>
> 1. While TreeDiffusion involves a slower inference process compared to TreeVAE, mainly due to the relatively slow denoising process and the need to traverse the tree from the root to the leaf, this is a tradeoff we willingly make. The slower inference and increased memory cost result in a significant improvement in cluster-specific image quality, with a marked increase in FID and visually clearer, more realistic images.
> 2. To generate a sample using TreeDiffusion, we first sample from the root of the learned latent tree and probabilistically traverse the tree to a leaf node. The path from the root to the selected leaf forms the conditioning signal. Starting from noise in image space, we denoise the image using the diffusion model conditioned on the latent features derived from the nodes along this path. No reference image is needed to extract these latent features.
> 3. After analyzing this issue, we found that this problem only occurred with the CUBICC dataset, likely due to its relatively small size. For all other datasets, the generated images from DiffuseVAE were much sharper, as expected.
> 4. Class-conditioning was not the focus of our paper. We concentrate on the unsupervised setting, where labels are not available. Therefore, leveraging clustering information, particularly from hierarchical clustering, provides an interesting and effective way to offer useful conditioning signals for training diffusion models for non-labeled data. However, we acknowledge that class conditioning would be an interesting baseline and will consider this suggestion for future work.

---

> > ### Comment · Reviewer_dZcu · 2024-11-25
> >
> > The authors’ response partially addresses my concerns but leaves critical issues unresolved. While they highlight the use of clustering information in an unsupervised setting, the effectiveness of TreeVAE clustering and TreeVAE-based conditioning remains unclear. To substantiate their claims, the authors should evaluate TreeVAE against alternative clustering methods to demonstrate:
> > 1. TreeVAE's effectiveness in producing meaningful clusters (why use TreeVAE for clustering?).
> > 2. The utility of TreeVAE-based conditioning in improving diffusion model performance (why TreeVAE helps diffusion performace while other cannot?).
> >
> > Currently, the evaluation is limited, as these aspects cannot be inferred from the provided experiments. While comparisons with DiffuseVAE and other baselines are planned for the final version, such evaluations are essential to establish the contribution of the method and should have been included earlier. Without clear comparisons or broader baselines, the overall contribution remains ambiguous, and the current evaluation does not adequately support the claims made.
> >
> > Overall, I decided to maintain my current rating.

---

### Official Review · Reviewer_GmBs · 2024-11-02

**Soundness:** 3
**Presentation:** 3
**Contribution:** 2
**Rating:** 5
**Confidence:** 3

**Summary:**

This study enhances the TreeVAE model for image generation by utilizing TreeVAE-generated clustering information as conditioning input for a diffusion model. The generated images were compared with those generated by TreeVAE, showing a quantified improvement.

**Strengths:**

The approach is conceptually straightforward and easy to understand. The paper is well-written.

Integrating clustering with conditional generation could have broad applications, particularly for datasets with distinct objects without labels.

**Weaknesses:**

Novelty: although adapting TreeVAE for image generation is intuitively reasonable with improvement, the contribution appears incremental, i.e., simply combining TreeVAE with a diffusion model.

Experiments: the ablation studies and comparisons are insufficient to substantiate the model's effectiveness. While the proposed TreeDiffusion model shows better quantitative generation metrics than TreeVAE, this improvement may largely stem from the addition of the diffusion model itself. I recommend that the authors isolate this effect. Specifically, quantitative metrics are not provided for image generation with a vanilla diffusion model, either in the model comparison or in the ablation section.

Further clarification of the above comment: In Table 2, more metrics related to image generation are suggested to be provided. Table 3 is expected to include additional metrics beyond FID and include results for a vanilla diffusion model for comparison.

**Questions:**

It is expected that the combination of clustering representations with a diffusion model holds the potential to advance learning performance for at least one of these two areas. However, the current approach falls short of maximizing this synergy, for example, concluding similar to [1] with a more robust integration. I also encourage the authors to include more related studies in this general sense within the related works section.

[1] Li T, Katabi D, He K. Self-conditioned image generation via generating representations, NeurIPS, 2024.

---

> ### Author Response · Authors · 2024-11-24
>
> We thank the reviewer for their thoughtful comments. Below, we address the identified weaknesses and questions.
>
> Regarding Novelty: \
> As explained in more detail in our General Response, the novelty of our work lies in conditioning the diffusion model on hierarchical information from TreeVAE. Unlike DiffuseVAE, which refines reconstructions and does not use latent representations for conditioning, our model solely relies on the conditional signal provided by the hierarchical latent information from TreeVAE. Furthermore, our model does not refine the image outputs from TreeVAE.
>
> Regarding the Experiments: \
> We would like to emphasize that the quantitative metrics for image generation with a vanilla diffusion model were provided in the text in Section 4.3. Nevertheless, we have included it in Table 3 for clarity and better comparison. Additionally, for the final version, we will try to expand Table 2 to include more metrics related to image generation to provide a more comprehensive evaluation.
>
> Regarding the last remark in the Questions: \
> Thank you for your suggestion! We have reviewed the mentioned work and included it in the related work section of the revised version of the paper.

---

### Official Review · Reviewer_d2pK · 2024-11-02

**Soundness:** 3
**Presentation:** 3
**Contribution:** 2
**Rating:** 3
**Confidence:** 4

**Summary:**

The paper introduces TreeDiffusion, an extension of TreeVAE which enhances its generation capabilities while maintaining its clustering performance. The major contribution of this work is to use diffusion models to generate samples using hierarchical structure learnt by TreeVAE. The authors also claim other minor novelties which are limited, e.g., they show that TreeDiffusion is able to generate cluster-specific samples; they replace the TreeVAE encoder-decoder architecture (which originally employed MLPs) with CNNs for better performance.

**Strengths:**

1. The paper is able to successfully overcome the limitations of  TreeVAE by using a diffusion model for generation. Particularly, the diffusion model is conditioned on the hierarchical structure learnt by TreeVAE to generate samples.

2. The experimental results show significantly better FID achieved by TreeDiffusion compared to its counterpart, TreeVAE.

**Weaknesses:**

1. I think the authors should refrain from calling the proposed method as an 'unified framework'. This is because the TreeVAE is trained independently of diffusion model, and the proposed method is inherently two-stage.

2. The novelty of the proposed method is very limited. Let me elaborate:
- There are two aspects of this work (a) hierarchical clustering and (b) conditional generation. The first part is taken as it is from TreeVAE with a minor architectural change (replacing the MLPs with CNN). Hence, the work should be judged on the basis of its novelty in conditional generation. For this part, the authors condition the reverse process on leaf-specific reconstruction and TreeVAE's hierarchical representation, $ p_\psi(x_{t-1}|x_t,\hat{x}_0^{(l)}, c_l) $ .

- Now, DiffusionVAE does a similar job by conditioning the reverse process on just the reconstruction  $p_\psi(x_{t-1}|x_t, \hat{x}_0)$. The difference however is that DiffusionVAE doesn't operate on TreeVAE, rather, it operates on vanilla VAE. From this, TreeDiffusion (current work) can be looked upon as a way to extend DiffusionVAE from vanilla VAE to TreeVAE. This extension is achieved by an additional condition over $c_l$. Doesn't this make TreeDiffusion a trivial extension of DiffusionVAE?

3. In the above light, TreeDiffusion is still refining the output of decoded leaf-node, similar to DiffusionVAE.

4. Although DiffuseVAE doesn't incorporate hierarchical structures, one can still make comparisons with DiffuseVAE in terms of FID in Table 1. I think this is important given the relation outlined in second point.

5. There is one confusion in the visual comparisons with TreeVAE. Are these images obtained using the MLP-based encoder-decoder architecture or the CNN-based encoder-decoder architecture?

6. I think one line of work that is not discussed in the paper but is closely related to unsupervised clustering is that of Unsupervised concept discovery  see [1,2,3]. I think it would be better to discuss these works, but comparison is not required.

7. Notationally, I would suggest the authors to use a different notation for hierarchical representations ($c_l$) which is similar to notation for decisions in the TreeVAE (line 168).

8. Authors follow the usual noising DDPM forward process. However, they alter the reverse process as shown in Eq. 7. Is this an ad-hoc change made by the authors? The reason for this question is that DDIM/DDPM make sure that marginal distribution remains the same as that is what is required in DDPM objective (cf. [4] for this). More specifically, DDIM and DDPM lead to same marginal despite having different joint forward distribution. Is there any such explanation for choice of Eq. 7?


Overall, I think that the paper is almost a trivial extension of DiffusionVAE. Hence I am resorting to a score of 3. However, I am ready to have a meaningful discussion with authors and am ready to alter the scores if the above points are appropriately addressed.


[1] Du, Yilun, et al. "Unsupervised learning of compositional energy concepts." _Advances in Neural Information Processing Systems_ 34 (2021): 15608-15620.

[2] Liu, Nan, et al. "Unsupervised compositional concepts discovery with text-to-image generative models."

[3] Su, Jocelin, et al. "Compositional image decomposition with diffusion models." _arXiv preprint arXiv:2406.19298_ (2024).

[4] Song, Jiaming, Chenlin Meng, and Stefano Ermon. "Denoising diffusion implicit models." _arXiv preprint arXiv:2010.02502_ (2020).

**Questions:**

See weaknesses.

---

> ### Author Response · Authors · 2024-11-24
>
> We thank the reviewer for their thoughtful comments. Below, we address the identified weaknesses and questions.
>
> 1. To clarify, we agree that our model is inherently two-stage and not trained end-to-end. However, when we refer to our method as a "unified framework," we aim to emphasize the seamless integration of the TreeVAE and diffusion model components. Specifically, our approach unifies the clustering and generative processes into one holistic generative clustering model. This integration enables the model to generate data specific to each learned cluster, providing a more effective and coherent solution that aligns cluster-specific data generation in a single pipeline.
>
> 2. As explained in more detail in our General Response, the novelty of our work lies in conditioning the diffusion model on hierarchical information from TreeVAE to improve cluster-specific data generation. Unlike DiffuseVAE, which refines the VAE reconstructions and does not condition on latent representations, we rely solely on latent hierarchical information. The authors of DiffuseVAE also experimented with conditioning on latent representations, which negatively impacted their performance in contrast to our approach.
>
> 3. As mentioned above, this is not the case. Unlike DiffuseVAE, TreeDiffusion does not refine the output of the decoded leaf node. Instead, we begin the denoising process from random noise and condition solely on the latent hierarchical information from TreeVAE, without using the decoded reconstructions. This key difference distinguishes our approach from DiffuseVAE.
>
> 4. Thank you for your suggestion. While we acknowledge the relevance of comparing with DiffuseVAE in terms of FID, we would like to highlight that our work already includes a comparison with a model that refines only the reconstructed outputs from TreeVAE during the reverse process, without conditioning on any latent information. This approach is conceptually similar to DiffuseVAE. Moreover, our ablation study demonstrates that our model, which conditions solely on the latent hierarchical information from TreeVAE—without utilizing the reconstructed images—outperforms this baseline. This result underscores the advantage of incorporating latent hierarchical information in our approach.
> For completeness, we will include a comparison with DiffuseVAE in terms of FID in the final version, using a vanilla VAE as described in the DiffuseVAE paper. However, it was not feasible to provide this comparison within the rebuttal period due to the significant time required to train the models.
>
> 5. The images in the visual comparisons with TreeVAE were obtained using the CNN-based encoder-decoder architecture.
>
> 6. We have reviewed the mentioned works and included them in the related work section of the revised version of the paper.
>
> 7. Thank you for pointing this out. We have adjusted the notation.
>
> 8. This is indeed an ad-hoc change introduced to speed up the inference process, thereby addressing the main limitation of DDPMs—namely, their slow sampling process.

---

> > ### Comment · Reviewer_d2pK · 2024-11-25
> >
> > I thank the authors for their detailed response. Yet my main concern regarding novelty in comparison with DiffuseVAE remains.
> >
> > > To clarify, we agree that our model is inherently two-stage and not trained end-to-end. However, when we refer to our method as a "unified framework," we aim to emphasize the seamless integration of the TreeVAE and diffusion model components. Specifically, our approach unifies the clustering and generative processes into one holistic generative clustering model. This integration enables the model to generate data specific to each learned cluster, providing a more effective and coherent solution that aligns cluster-specific data generation in a single pipeline.
> >
> > I understand this point. Maybe it would be better to write this explicit.
> >
> > > As explained in more detail in our General Response, the novelty of our work lies in conditioning the diffusion model on hierarchical information from TreeVAE to improve cluster-specific data generation. Unlike DiffuseVAE, which refines the VAE reconstructions and does not condition on latent representations, we rely solely on latent hierarchical information. The authors of DiffuseVAE also experimented with conditioning on latent representations, which negatively impacted their performance in contrast to our approach.
> >
> > > As mentioned above, this is not the case. Unlike DiffuseVAE, TreeDiffusion does not refine the output of the decoded leaf node. Instead, we begin the denoising process from random noise and condition solely on the latent hierarchical information from TreeVAE, without using the decoded reconstructions. This key difference distinguishes our approach from DiffuseVAE.
> >
> > I am not convinced on this point. I agree that the *pitch* of DiffuseVAE is to refine the VAE reconstruction. Whereas the the *pitch* of current work is to improve cluster specific data generation. I agree that both methods come out as a solution to their own problem. However, from mathematical and implementation perspective, there is very minimal change. While TreeDiffusion learns , $p_\psi(x_{t-1}|x_t, \hat{x}_0^{(l)}, c_l)$,
> >
> > DiffuseVAE learns  $p_\psi(x_{t-1}|x_t, \hat{x}_0)$. From this perspective, the only difference between DiffuseVAE and TreeDiffusion is that of incorporating the additional conditioning $c_l$. One can argue that TreeDiffusion is DiffuseVAE (on TreeVAE) + conditioning on hierarchical information, $c_l$. I would like to hear author's thoughts on this.
> >
> > > Thank you for your suggestion. While we acknowledge the relevance of comparing with DiffuseVAE in terms of FID, we would like to highlight that our work already includes a comparison with a model that refines only the reconstructed outputs from TreeVAE during the reverse process, without conditioning on any latent information. This approach is conceptually similar to DiffuseVAE. Moreover, our ablation study demonstrates that our model, which conditions solely on the latent hierarchical information from TreeVAE—without utilizing the reconstructed images—outperforms this baseline. This result underscores the advantage of incorporating latent hierarchical information in our approach. For completeness, we will include a comparison with DiffuseVAE in terms of FID in the final version, using a vanilla VAE as described in the DiffuseVAE paper. However, it was not feasible to provide this comparison within the rebuttal period due to the significant time required to train the models.
> >
> > Thanks for this clarification. When you say '*we would like to highlight that our work already includes a comparison with a model that refines only the reconstructed outputs from TreeVAE during the reverse process, without conditioning on any latent information. This approach is conceptually similar to DiffuseVAE*', doesn't this prove the point that I outlined above?
> >
> > > The images in the visual comparisons with TreeVAE were obtained using the CNN-based encoder-decoder architecture.
> >
> > > We have reviewed the mentioned works and included them in the related work section of the revised version of the paper.
> >
> > > Thank you for pointing this out. We have adjusted the notation.
> >
> > > This is indeed an ad-hoc change introduced to speed up the inference process, thereby addressing the main limitation of DDPMs—namely, their slow sampling process.
> >
> > Thanks for these clarifications. It would be better to mention that you have used CNN architecture for the visual results.

---

### Meta-Review · Area_Chair_gB4V · 2024-12-20

**Metareview:**

This work presents TreeDiffusion, a two-stage framework that integrates a hierarchical clustering model with a diffusion model to enhance generation quality. TreeDiffusion first learns a hierarchical latent tree and then parameterizes a diffusion model conditioned on raw predictions sampled from the latent tree. While the paper is well-motivated, its primary limitation is the incremental nature of its contribution. All reviewers raised concerns about the novelty of the approach, which the authors were unable to fully address during the rebuttal process. Consequently, the AC recommends rejection.

**Additional Comments On Reviewer Discussion:**

Reviewer d2pK's primary concern is the limited novelty of the proposed method. Reviewer d2pK believes the model comprises two components: (a) hierarchical clustering and (b) conditional generation. The hierarchical clustering is derived from TreeVAE, while the conditional generation conditions the reverse process on leaf-specific reconstruction and TreeVAE's hierarchical representation. Additionally, the approach shows similarities to DiffusionVAE. The authors failed to address these concerns regarding novelty in the rebuttal, leading Reviewer d2pK to recommend rejecting the paper.

Reviewers GmBs, dZcu, and E4q2 shared similar concerns about the limited novelty and highlighted insufficient experimental validation. The authors were unable to alleviate these concerns, and all reviewers ultimately leaned toward rejecting the paper due to its lack of novelty and inadequate experimental support.

---

### Decision · Program_Chairs · 2025-01-22

Reject